# Functional analysis in a model sea anemone reveals phylogenetic complexity and a role in cnidocyte discharge of DEG/ENaC ion channels

Jose Maria Aguilar-Camacho[1,3,4], Katharina Foreman[2,4], Adrian Jaimes-Becerra[1], Reuven Aharoni[1], Stefan Gründer [2✉] & Yehu Moran [1✉]

Ion channels of the DEG/ENaC family share a similar structure but serve strikingly diverse biological functions, such as $Na^+$ reabsorption, mechanosensing, proton-sensing, chemo-sensing and cell-cell communication via neuropeptides. This functional diversity raises the question of the ancient function of DEG/ENaCs. Using an extensive phylogenetic analysis across many different animal groups, we found a surprising diversity of DEG/ENaCs already in Cnidaria (corals, sea anemones, hydroids and jellyfish). Using a combination of gene expression analysis, electrophysiological and functional studies combined with pharmacological inhibition as well as genetic knockout in the model cnidarian *Nematostella vectensis*, we reveal an unanticipated role for a proton-sensitive DEG/ENaC in discharge of *N. vectensis* cnidocytes, the stinging cells typifying all cnidarians. Our study supports the view that DEG/ENaCs are versatile channels that have been co-opted for diverse functions since their early occurrence in animals and that respond to simple and ancient stimuli, such as omnipresent protons.

[1] Department of Ecology, Evolution and Behavior, Alexander Silberman Institute of Life Sciences, Faculty of Science, The Hebrew University of Jerusalem, Jerusalem, Israel. [2] Institute of Physiology, RWTH Aachen University, Aachen, Germany. [3] Present address: Department of Biological Sciences, North Carolina State University, Raleigh, NC, USA. [4] These authors contributed equally: Jose Maria Aguilar-Camacho, Katharina Foreman. ✉email: sgruender@ukaachen.de; yehu.moran@mail.huji.ac.il

Members of the degenerin/epithelial Na$^+$ channel (DEG/ENaC) superfamily share a similar structure with two transmembrane domains (TMDs), a large extracellular domain (ECD), and cytosolic amino- and carboxy-termini[1]. A functional DEG/ENaC ion channel is a complex of three subunits[1,2]. Despite these common structural features and their common ancestry, and in striking contrast to many other ion channel families, members of the DEG/ENaC superfamily are highly diverse in function and activation mechanisms, both in a given phylum and between phyla[3,4].

In mammals, there are typically nine different genes coding for DEG/ENaCs, which include the epithelial Na$^+$ channels (ENaCs), acid-sensing ion channels (ASICs) and the bile acid-sensitive ion channel (BASIC)[4]. ENaCs are constitutively open channels that contribute to whole-body Na$^+$ homeostasis[5]. ASICs are gated by extracellular protons and modulated by extracellular Ca$^{2+}$ ions and/or neuropeptides[6,7]. They are expressed in the central and peripheral nervous system where they act as proton sensors[6,8]. BASIC is gated by bile acids and blocked by extracellular Ca$^{2+}$ ions[9]. It is found in the epithelial cells of the bile ducts, but its physiological function remains uncertain[10]. While ENaC evolved early in vertebrates and might have aided in the osmoregulatory adaption during the transition of vertebrates to land[11], proton-sensitive ASICs are also present in non-vertebrate deuterostomes and some protostomes and likely evolved before the Cambrian explosion[12,13]. Although ENaC and ASICs are not close relatives within the DEG/ENaC family[14,15], it has been proposed that ENaC evolved from the ASIC lineage[11].

In insects, a large family of DEG/ENaCs called Pickpockets (PPKs) was studied mostly in the fly *Drosophila melanogaster* that carries 31 genes of this family[16]. They are expressed in glia and selected neuronal subpopulations, responding to mechanical, thermal, and chemical stimuli[17], and mediating diverse functions such as locomotion[18] and salt and water taste[17,19]. The nematode *Caenorhabditis elegans* has 30 DEG/ENaC genes called *degenerins*, some of which are found in touch receptor neurons where they are gated by touch stimuli[20,21]. In snails, a DEG/ENaC expressed in neurons is called FaNaC and is directly gated by the neuropeptide FMRFamide[22]. In the marine annelid *Platynereis dumerilii*, many DEG/ENaCs were identified, and one of them, called MGIC, is broadly expressed in the larval brain and is gated by a group of related neuropeptides, the Wamides[23]. FaNaC and MGIC indeed belong to a broader subfamily of peptide-gated DEG/ENaCs that are found in several lophotrochozoans[24].

The phylum Cnidaria, which includes sea anemones, corals, jellyfish, and hydroids, is a sister group to Bilateria and represents a relatively basal branch within Metazoa. It can be further divided in two subphyla: Anthozoa (sea anemones and corals) and Medusozoa (jellyfish and hydroids)[25]. Twelve DEG/ENaCs were identified in the hydroid freshwater polyp *Hydra magnipapillata* (subphylum Medusozoa), the HyNaCs. HyNaCs assemble into a variety of heterotrimers that are uniformly gated by just two *Hydra* neuropeptides, Hydra-RFamide 1 and Hydra-RFamide 2[14,26]; the only exception is HyNaC12, for which the activating stimulus is unknown[14]. HyNaCs are differentially expressed in the peduncle and at the base of the tentacles, potentially playing a role in muscle contraction and feeding behavior[14,27,28].

Finally, while DEG/ENaCs are absent from most unicellular eukaryotes, with exceptions in the clades Heterokonta and Filasterea[29], they are also present in all other groups of basal animals: sponges, ctenophores[30], and the placozoan *Trichoplax adhaerens*[15], demonstrating that DEG/ENaCs emerged at the base of the animal tree of life. In *T. adhaerens*, 11 genes were discovered, called TadNaCs. One of them, TadNaC6, is constitutively open, modulated by alkaline pH and blocked by external H$^+$ and Ca$^{2+}$ ions[15].

In summary, DEG/ENaCs have very diverse activation stimuli and their variety in different animal groups is striking. This raises the question of the ancient properties of DEG/ENaCs. Because nearly all DEG/ENaCs of *Hydra* are activated by neuropeptides and because different lophotrochozoan species also have peptide-gated DEG/ENaCs, it has been proposed that DEG/ENaCs evolved from a peptide-gated channel[14]. However, it cannot be excluded that *Hydra* lost genes for other ancestral DEG/ENaCs with different gating mechanisms.

To further resolve the open questions of the evolution of DEG/ENaCs, we turned to the sea anemone *Nematostella vectensis*, a member of the cnidarian subphylum Anthozoa, which branched from the Medusozoan subphylum >500 million years ago[31]. We identified 29 DEG/ENaCs, which we named *Nematostella* Sodium Channels (NeNaCs). The sequences of NeNaCs are surprisingly diverse, suggesting an extensive diversification of DEG/ENaCs in *Nematostella*. Employing molecular, genetic and electrophysiological methods, we characterized one NeNaC in detail, revealing a proton-sensitive DEG/ENaC that mediates discharge of cnidocytes. Thus, our study reveals a complex evolution of the DEG/ENaC superfamily that is characterized by extensive phylum-specific gene losses and duplications. In addition, we identify an ion channel that mediates cnidocyte discharge by the omnipresent ligand protons.

## Results

**Phylogenetic analysis reveals the complexity of DEG/ENaCs.** As a first step, we searched for sequences encoding DEG/ENaCs in publicly available genomic and transcriptomic databases. Our analysis revealed that the DEG/ENaC superfamily greatly expanded and diversified in the phylum Cnidaria: many homologs were identified in the available genomes of different species belonging to the subphyla Anthozoa and Medusozoa: 29 genes in *N. vectensis* (NeNaCs), 16 in *Scolanthus callimorphus*, 21 in *Pocillipora damicornis*, 19 in *Actinia tenebrosa*, 14 in *Amplexidiscus fenestrafer*, 26 in *Stylophora pistillata*, 37 in *Clytia hemisphaerica*, 11 in *H. magnipapillata* (HyNaCs) and 18 in *Aurelia aurita* (Supplementary Data 1). In addition, we retrieved DEG/ENaCs from the marine sponge *Amphimedon queenslandica* and from the comb jelly *Mnemiopsis leidyi*. Due to the uncertainty around the root of the animal tree as both sponges and ctenophores have been recovered as the sister group to all other animals in different studies[32–34], we performed the phylogenetic analyses using both taxa as outgroups in distinct analyses. The results of our likelihood and Bayesian phylogenetic analyses when channels from *A. queenslandica* were placed as outgroup are largely congruent with each other (Fig. 1a and Supplementary Figs. 1–3). Two major molecular clades were discovered, which both contain cnidarian DEG/ENaCs: Clade A contains in addition deuterostome ASICs, vertebrate BASIC, all PPKs, some degenerins, and most TadNaCs, and Clade B contains, in addition, human ENaCs, FaNaCs/WaNaCs from annelids, additional degenerins, and one TadNaC (TadNaC10).

The sequences of Cnidaria are polyphyletic and dispersed in six different molecular subclades within the two major clades: subclades I, II, III, and IV in clade A; subclades V and VI in clade B. Subclade I contains a monophyletic branch with members of Anthozoa (i.e., NeNaC2) and this subclade is part of a monophyletic subclade with BASIC and ASICs from deuterostomes, two Degenerins (Del9 and Del10), most TadNaCs, and some additional sequences of Anthozoa (i.e., NeNaC23 and *A. tenebrosa* 7) (Supplementary Figs. 1–3). Subclade II contains only sequences from Cnidaria (Anthozoa and Medusozoa, for example HyNaCs, NeNaC1, and NeNaC24). Subclade III also contains only sequences of Anthozoa (i.e., NeNaC6,

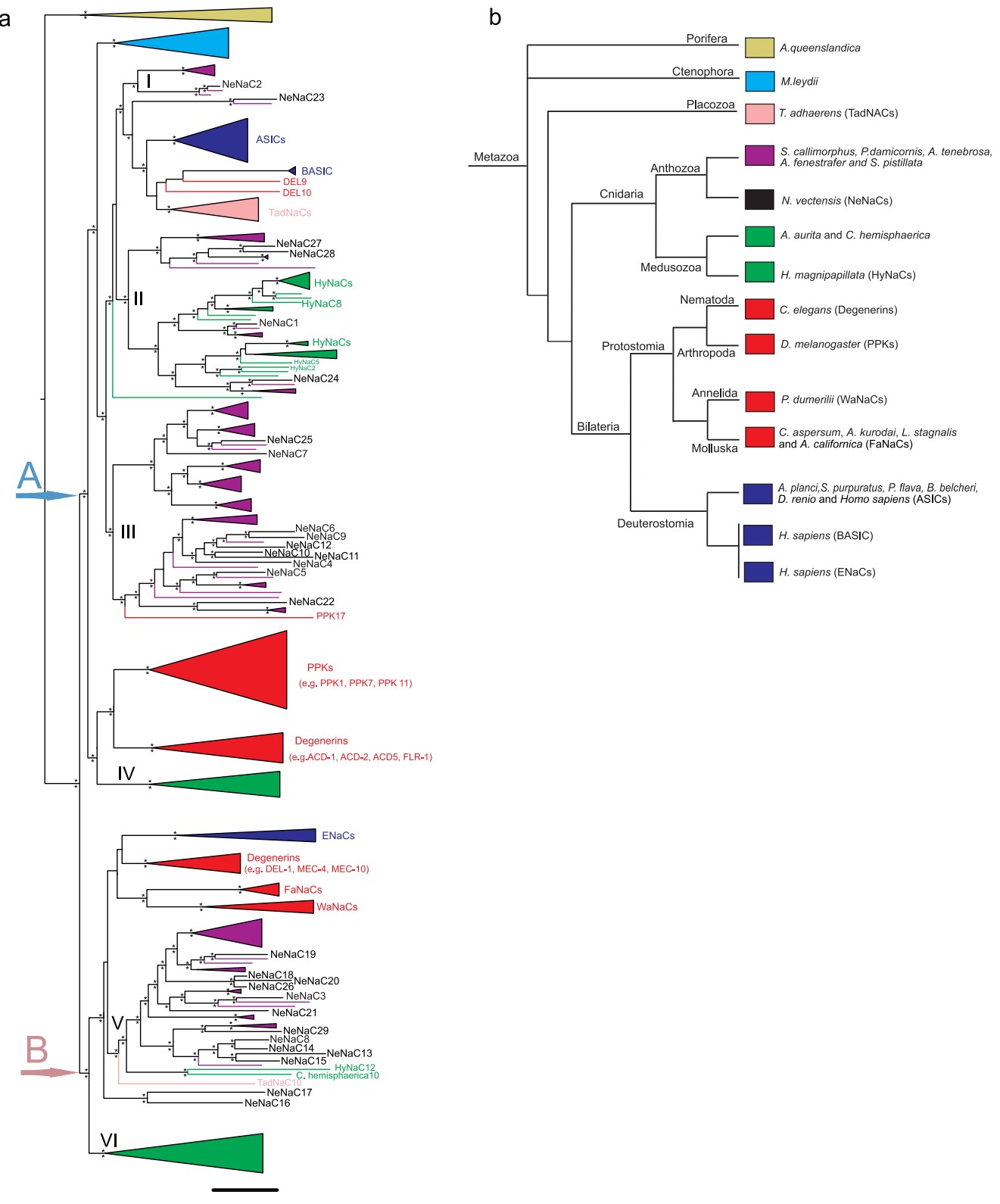

**Fig. 1 Molecular phylogenetic tree of the DEG/ENaC channel superfamily with *A. queenslandica* as an outgroup. a** The tree was constructed with the IQ-Tree software (see Materials and Methods). The branches of some clades are collapsed. Asterisks above and below branches denote a support value > 65% for the three different methods calculated in IQTREE and bayesian posterior probabilities >0.65 calculated in phylobayes, respectively. **b** Schematic representation of the phylogenetic relationship of the main groups of species selected for the sampling of their DEG/ENaCs.

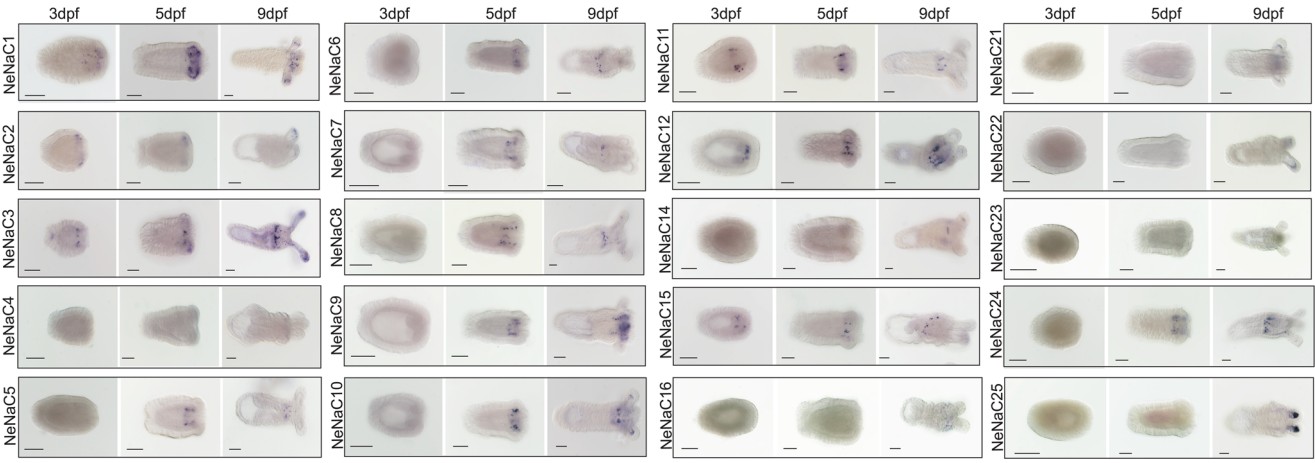

**Fig. 2 In situ Hybridization of selected NeNaCs at three different developmental stages in _N. vectensis_.** The expression is noticeable by the blue stain (NBT/BCIP crystals). Oral end of the animals is to the right for each panel. dpf days post-fertilization. Scale bars: 100 µm.

NeNaC9) with the striking exception of one sequence of _D. melanogaster_, PPK17, the sole member of a subfamily of PPKs[16]. With sponges as an outgroup, sequences of Ctenophora are monophyletic and a sister clade to subclades I-III. Subclade IV contains some sequences of Medusozoa (only of _C. haemisphaerica_) and this is a sister clade to most PPKs and some degenerins (i.e., ACD-1, ACD-5). Subclade V of clade B contains only cnidarian DEG/ENaCs, mostly of Anthozoa, although two sequences of Medusozoa (HyNaC12 and _Clytia haemispherica_10) are within this subclade. Subclade V is part of a monophyletic subclade containing human ENaCs, TadNaC10, additional Degenerins (i.e., DEL-1, MEC-4, MEC-10), FaNaCs, and WaNaCs from Annelida (for example peptide-gated MGIC) and two additional NeNaCs (NeNaC16 and NeNaC17). It is a sister clade to subclade VI containing sequences of Medusozoa (excluding HyNaCs).

If the trees are rooted with Ctenophora channels, however, DEG/ENaCs from the sponge _A. queenslandica_ become monophyletic and a sister clade to the major clade B (Supplementary Figs. 4–7). Despite the robustness between the different phylogenetic analyses we present in the current study, we cannot exclude the possibility that future studies on the evolution of this protein superfamily utilizing a larger sampling can establish a broader picture of the phylogenetic relationships of the different channel families.

**Spatiotemporal expression patterns of DEG/ENaCs of _N. vectensis_.** We uncovered the expression patterns of 20 NeNaCs (out of 29) at three different developmental stages, early planula (3 days post-fertilization, dpf), metamorphic phase (5 dpf), and primary polyp (9 dpf) using in situ hybridization (ISH) (Fig. 2). No expression was detected for any of the NeNaCs in the gastrula stage (1 dpf). Three NeNaCs showed a unique expression pattern at their three developmental stages: NeNaC1, NeNaC2, and NeNaC3. NeNaC1 is expressed in endodermal cells and a few ectodermal cells in the oral end in the planula and metamorphic phase. In the primary polyp, the expression is mostly in endodermal cells of the tentacles and pharynx (Fig. 2; Supplementary Fig. 8a, b). These cells could be tentacular neurons, as in previously published single-cell RNA sequencing (scRNA-seq) data this gene was upregulated in a "metacell" of neurons[35]. NeNaC2 is expressed in ectodermal cells at the oral end in the planula and in the metamorphic phase; and in the primary polyp the expression is in ectodermal cells of the tentacles (Fig. 2; Supplementary Fig. 8c, d), a body region known to be rich in cnidocytes,

the unique stinging cells of cnidarians. The identity of these cells as cnidocytes is further supported by scRNA-seq data that revealed upregulation of this gene in the "metacell" of cnidocytes[35].

NeNaC3 is expressed at the developing pharynx and at a domain of the aboral end in the planula. In the metamorphic phase, the expression is in endodermal and ectodermal cells in the developing pharynx and in the oral end. In the primary polyp, the expression is in endodermal cells in the pharynx and in ectodermal cells of the tentacles. For the vast majority of NeNaCs (e.g., NeNaC6, NeNaC7, NeNaC24), the expression pattern is in endodermal cells of the developing pharynx in the planula and in the metamorphic phase. In the primary polyp, the expression is in endodermal cells of the pharynx. NeNaC8 and NeNaC10 are expressed in endodermal cells of the tentacles and pharynx in the primary polyp. NeNaC21, NeNac22, and NeNaC25 are expressed in endodermal cells of the tentacles in the primary polyp and NeNaC15 is expressed in endodermal cells of the mesenteries in the primary polyp (Fig. 2).

**Some NeNaCs form functional homomers in _Xenopus_ oocytes.** We selected 15 NeNaCs with a clear expression in ISH for functional analysis (NeNaC1-NeNaC3, NeNaC5-NeNaC12, NeNaC14, NeNaC15, NeNaC23, and NeNaC24). The length of these 15 NeNaCs varied from 458 to 578 amino acids (Supplementary Fig. 9). Typical hallmarks of DEG/ENaCs were completely conserved (Supplementary Fig. 9): two putative TMDs with a HG motif proximal of TMD1, a Trp residue at the beginning of TMD1, 11-14 conserved Cys residues in the ECD, and a GxS motif in TMD2, which is crucial for ion selectivity[36,37]. The conservation of these hallmarks suggests that NeNaCs form functional cation channels.

To functionally characterize these 15 NeNaCs, we expressed them in _Xenopus laevis_ oocytes and performed two-electrode voltage clamp (TEVC). We started by co-injecting up to nine different NeNaCs that belong to the same clade (A or B); subsequently, we injected increasingly smaller groups and different combinations until we injected single NeNaCs. Based on the known properties of DEG/ENaCs from other species, we tested for proton-sensitivity by applying an acidic bath solution (pH 4.0) and for constitutively active channels by applying the pore blockers amiloride (100 µM) or diminazene (10 µM). In addition, we tried to unspecifically activate NeNaCs by applying a bath solution with a low concentration of divalent cations (10 µM $Ca^{2+}$, no $Mg^{2+}$). Not all stimuli were applied to all NeNaCs

(Supplementary Table 1). This analysis revealed that NeNaC2 and NeNaC14 were sensitive to protons and that NeNaC8-expressing oocytes had small constitutive currents that were sensitive to amiloride (100 μM) and diminazene (10 μM). In addition, in oocytes that expressed NeNaC8, reducing $[Ca^{2+}]$ in the bath solution elicited large currents. These results suggest that these three NeNaCs form functional homomeric channels in *Xenopus* oocytes.

To obtain further information on these NeNaCs, we mutated the degenerin (DEG) position, which is at the extracellular end of TMD2. NeNaCs have amino acids with a small side chain at this position (Gly, Ala or Ser; Supplementary Fig. 9). Introducing an amino acid with a large side chain, such as threonine (Thr) constitutively opens some DEG/ENaCs, allowing to detect channel activity in the absence of a physiological ligand. We again co-injected up to ten different NeNaCs with the DEG mutation (DEG-NeNaCs) that belong to the same clade and then injected increasingly smaller groups until we injected single DEG-NeNaCs. We applied the same stimuli as for wt channels (pH 4.0, amiloride, diminazene, low divalent cations). Although we did not comprehensively test all possible combinations, large constitutive currents (up to ~15 μA), which were sensitive to amiloride and diminazene, were observed whenever a pool contained either NeNaC1-G495T or NeNaC2-G516T from clade A, or NeNaC8-G423T, NeNaC14-G471T or NeNaC15-G460T from clade B. These channels also gave rise to constitutive currents when injected alone and, in addition, were also sensitive to protons (Supplementary Fig. 10). NeNaC3-S464T was not constitutively active, but protons (pH 5 and pH 6.0) elicited small currents (up to 1 μA). These findings confirm that NeNaC2, NeNaC8, and NeNaC14 form homomeric channels in oocytes. In addition, they reveal that NeNaC1, NeNaC3, and NeNaC15 form homomeric channels as well. In summary, we obtained evidence that six NeNaCs can form homomeric channels in *Xenopus* ooyctes. From our analysis of oocytes co-expressing more than one NeNaC or DEG-NeNaC, we obtained no evidence for specific heteromeric channels. Nevertheless, a more systematic analysis is required to clearly reveal the formation of heteromeric NeNaCs.

To test for a peptide-gated ion channel among those NeNaCs, we applied 27 different *Nematostella* neuropeptides[38–40] (1–30 μM; Supplementary Table 2). However, none of these peptides elicited currents in oocytes expressing different combinations of NeNaCs or individual NeNaCs (Supplementary Table 2).

**NeNaC2 is a proton-sensitive Na$^+$ channel**. We analyzed the functional properties of proton-sensitive NeNaC2 and NeNaC14 and of constitutively active NeNaC8 in more detail. Proton-sensitivity of NeNaC2 was determined by applying increasingly more acidic pH levels ranging from pH 7.0 to pH 4.0, revealing half-maximal activation at pH $5.8 \pm 0.7$ ($n = 9$; Fig. 3a, b). Current amplitude saturated at about pH 5.5 and up to this pH, currents showed no strong desensitization. At pH values below pH 5.5, however, currents became biphasic, with a brief transient component followed by a sustained component. The current amplitude of the sustained component successively decreased at more acidic pH (Fig. 3a).

HyNaCs have a high $Ca^{2+}$ permeability[41], leading to biphasic current in oocytes: the transient part due to secondary activation of $Ca^{2+}$-activated $Cl^-$ Channels (CaCCs), which are endogenous to *Xenopus laevis* oocytes, and the sustained part due to influx via HyNaCs. Chelating $Ca^{2+}$ by injecting the $Ca^{2+}$ chelator EGTA into oocytes efficiently inhibits activation of CaCCs and abolishes the fast-transient component of the current[41]. To test whether the biphasic NeNaC2 currents were due to $Ca^{2+}$ permeability and

activation of the endogenous CaCC, we, therefore, injected NeNaC2-expressing oocytes with EGTA. However, EGTA injection did not abolish the biphasic current (Fig. 3c). Thus, it appears that NeNaC2 does not conduct substantial amounts of $Ca^{2+}$ and that it only desensitizes at strong acidic pulses, but not at more physiological pH values, where it carries simple on-off currents. We assessed ion selectivity of NeNaC2 more systematically and determined the reversal potential $E_{rev}$ with Na$^+$, K$^+$, and $Ca^{2+}$ as the main extracellular cation (Fig. 3d, e). This revealed a $P_{Na}/P_K = 4.9$ and $P_{Na}/P_{Ca} = 4.8$. In summary, NeNaC2 is a moderately selective Na$^+$ channel.

Next, we determined sensitivity of NeNaC2 to amiloride and diminazene at pH 6.0. Amiloride inhibited NeNaC2 with a low apparent affinity ($IC_{50} = 1.3 \pm 2.3$ mM; $n = 10$; Fig. 3f, g); even at high amiloride concentrations, the inhibition was incomplete. Diminazene, in contrast, had a higher potency with an $IC_{50}$ of $2.4 \pm 0.9$ μM ($n = 14$) and complete inhibition at high concentrations (Fig. 3f, g).

Previous work on ASICs revealed that a highly conserved histidine residue or a pair of histidine residues, just distal to TMD1 at the beginning of beta-sheet 1 (β1), is indispensable for proton-sensitivity of ASICs[12,42–44]. Sequence alignment of NeNaC2 with ASIC1a, ASIC2, and ASIC3 revealed that NeNaC2 has a histidine at position 141, two positions proximal of the critical histidine residue in ASICs (Fig. 3i). While substitution of this histidine residue in NeNaC2 to alanine (H141A) reduced proton-evoked currents, it did not reduce apparent pH sensitivity (Fig. 3h, j), demonstrating that H141 is not essential for proton sensitivity of NeNaC2. Future studies need to clarify the reason for reduction in current amplitude and the exact role, if any, of H141 for proton-gating of NeNaC2.

Proton-sensitivity of NeNaC14 was substantially lower than that of NeNaC2 with half-maximal activation at pH < 4.0 ($n = 11$; Fig. 4a, b). We, therefore, did not analyze the properties of NeNaC14 in more detail. We determined the current amplitude of NeNaC8-expressing oocytes with different concentrations of extracellular $Ca^{2+}$ (no $Mg^{2+}$), revealing an apparent $IC_{50}$ for $Ca^{2+}$ of $20 \pm 30$ μM ($n = 15$; Fig. 4c, d). Thus, activity of NeNaC8 is tightly regulated by $[Ca^{2+}]_e$, similar to rat BASIC[45]. Because the $Ca^{2+}$ concentration needs to be reduced to unphysiologically low levels to activate NeNaC8, it is highly unlikely that this is the physiological stimulus to open NeNaC8. We determined the apparent affinity of NeNaC8 to amiloride in a bath solution containing 100 nM $Ca^{2+}$, and apparent affinity to diminazene in a bath solution containing 1.8 mM or 100 nM $Ca^{2+}$. Apparent $IC_{50}$ of amiloride was $310 \pm 180$ μM ($n = 11$; Fig. 4e, f) and apparent $IC_{50}$ of diminazene $124 \pm 75$ nM ($n = 14$; 1.8 mM $Ca^{2+}$) and $250 \pm 230$ nM ($n = 14$; 0.1μM $Ca^{2+}$; Fig. 4g–i), suggesting that diminazene does not compete with $Ca^{2+}$.

**NeNaC2 mediates acid-induced cnidocyte discharge**. Proton sensitivity of NeNaC2 to pH values in the physiological range was particularly intriguing. Therefore, and because NeNaC2 is expressed in cnidocytes (Fig. 2; Supplementary Fig. 8c, d), we tested whether NeNaC2 plays a functional role in cnidocytes. First, we assessed the capture of brine shrimp nauplii by 3 months old *Nematostella* polyps in the presence and absence of the NeNaC2 inhibitor diminazene. We found that diminazene (100 μM) delayed the capture in comparison to untreated polyps (Fig. 5a–d). We then assessed the discharge of cnidocysts, the typifying explosive organelles found inside cnidocytes, at nearly neutral pH 7.2 and at acidic pH. In the presence of prey extract, cnidocysts were similarly discharged at pH 7.2 and at acidic pH (pH 6.0 or 5.5) (Fig. 5e–g and Supplementary Fig. 11a, b). Diminazene slightly reduced the cnidocyst discharge induced by

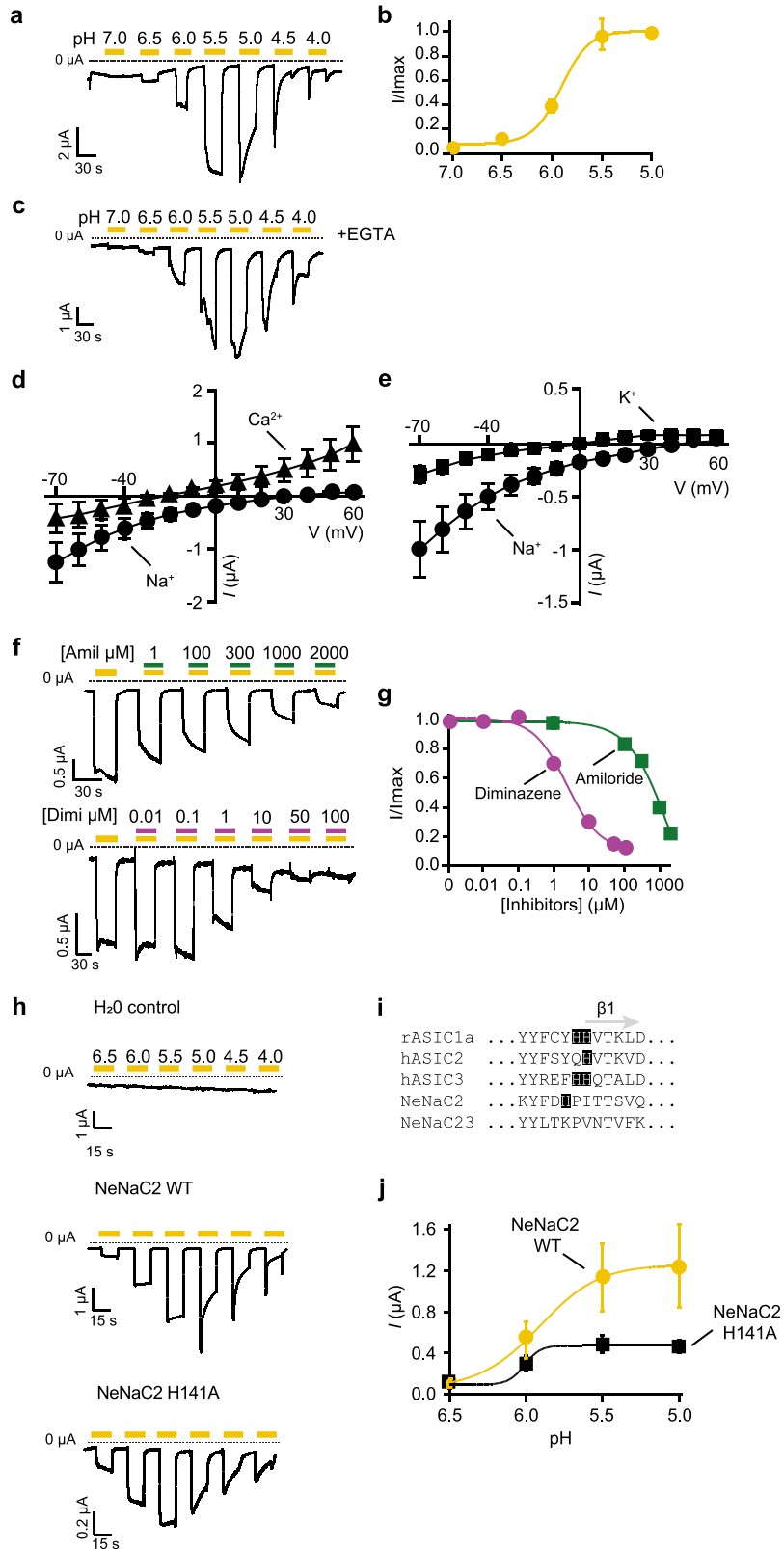

prey extract at pH 7.2 and pH 6.0, but this reduction was not significant ($p > 0.05$). At pH 5.5, however, diminazene significantly reduced cnidocyst discharge ($p < 0.05$; Supplementary Fig. 11a, b). In the absence of prey extract, cnidocysts did not discharge at pH 7.2. However, as previously observed[46], acidic pH induced the discharge of the cnidocysts even in the absence of prey extract. Strikingly, this acid-induced cnidocyst discharge was

completely abolished by diminazene (Fig. 5e–g and Supplementary Fig. 11a, b) suggesting that NeNaC2 might mediate the acid-induced cnidocyte discharge. To gain further insight into the functional role of NeNaC2 in *Nematostella* we generated a null mutant for the gene encoding this channel. These NeNaC2 null homozygous mutants (NeNaC2$^{(-/-)}$) had a deletion of five nucleotides (Fig. 5h), causing a frameshift at the amino acid level

**Fig. 3 NeNaC2 is a proton-gated ion channel. a** Representative trace of NeNaC2 currents elicited by increasingly more acidic pH (yellow bars). Note the biphasic current that appeared below pH 5.5. *Right.* **b** Concentration–response curve ($n = 9$). **c** Representative trace from NeNaC2-expressing oocytes injected with EGTA. **d** I/V plot revealing $E_{rev} = +46$ mV for $Na^+$ (black circles) and $E_{rev} = -30$ mV for $Ca^{2+}$ (black triangles). **e** I/V plot revealing $E_{rev} = +46$ mV for $Na^+$ (black circles) and $E_{rev} = +6$ mV for $K^+$ (black circles). *Error bars* represent S.D. **f** Representative current traces of NeNaC2 in the presence of increasing concentrations of amiloride or diminazene (green and turquoise bars, concentrations in µM); NeNaC2 was activated with pH 6.0. **g** Concentration–response curves. *Error bars* represent S.D. **h** Representative current traces of NeNaC2 WT, NeNaC2H141A, and of water-injected control oocytes. **i** Alignment of the region close to β1 of rASIC1a, hASIC2, hASIC3, NeNaC2, and NeNaC23. **j** Mean current amplitudes of NeNaC2 (yellow dots, $n = 8$) and NeNaC2H141A (black squares, $n = 6$). Current amplitudes were reduced for the mutant, but apparent proton affinity was similar. *Error bars* represent S.E. Concentration–response curves were fitted with the *Hill* equation.

affecting the proper formation of a functional homomeric channel (Supplementary Table 3) as demonstrated by western blot with a specific NeNaC2 antibody (Fig. 5i). The prey capture experiment was carried out for the F2 progeny of NeNaC2, and no significant differences were detected in the pairwise comparisons between the three genetic pools (NeNaC2(+/+), NeNaC2(+/−) and NeNaC2(−/−)) during the capture of the 1st, 2nd and 3rd artemia (Supplementary Fig. 12a,b). As described above, at acidic pH (6.0), NeNaC2$^{(+/+)}$ wild-type polyps discharged cnidocysts in the absence of prey extract. NeNaC2$^{(+/−)}$ heterozygous and NeNaC2$^{(−/−)}$ null homozygous mutants, however, discharged a significantly reduced number of cnidocysts compared to NeNaC2 $^{(+/+)}$ wild-type organisms. This reduction in cnidocyst discharge was particularly pronounced for NeNaC2$^{(−/−)}$ null homozygous mutants (Fig. 5j, k). Together, these results provide compelling evidence that NeNaC2 mediates acid-induced cnidocyte discharge.

## Discussion

Our study has two major findings: it revealed that there is a greater diversification of the DEG/ENaC channels in Cnidaria than was previously recognized, and it uncovered a proton-sensitive DEG/ENaC in *N. vectensis* that is not directly related to the subfamily of deuterostome ASICs and mediates acid-induced cnidocyte discharge.

The different paralog copies of selected cnidarian species cluster in six distinct subclades within the phylogenetic tree presented here (Fig. 1a). Most notably, cnidarian DEG/ENaCs are present in major clades A and B, highlighting that the common ancestor of cnidarian species already contained representatives from these two clades. These findings are in agreement with a recent study[29], which included a phylogenetic analysis of DEG/ENaCs. This study recovered two major clades (ASIC and ENaC superclusters) within the larger DEG/ENaC family that are of ancient pre-bilaterian origin. These major clades are congruent with ours (clades A and B), however, there are also differences, possibly resulting from the other study using PPK channels as their outgroup. *H. magnipapillata* has 12 DEG/ENaCs, which is a small number of genes compared to other marine and brackish cnidarian species (such as *N. vectensis*). Moreover, all HyNaCs cluster in subclade II of clade A, except for HyNaC12 which is in subclade V of clade B (Fig. 1a). Thus, the analysis of HyNaCs vastly underestimated the variety of cnidarian DEG/ENaCs. Notably, even other hydrozoans such as *C. hemisphaerica* carry many more DEG/ENaCs (37 copies) which occupy more diverse positions on the phylogenetic tree (Fig. 1a). Thus, the *Hydra* lineage either lost many DEG/ENaCs that were present at the base of the cnidarian branch or other cnidarians diversified their DEG/ENaCs more extensively than *Hydra*. The reduced complexity of DEG/ENaCs in *Hydra* may be related to the freshwater habitat of *Hydra* because a similar situation is found in the phylum Porifera, where ion channels from different families (including DEG/ENaCs) are absent in freshwater species (Class: Demospongiae), based on the genome and transcriptome sequences, in comparison to selected marine sponge species (i.e.,

*A. queenslandica*) belonging to the same class where they are present[47,48].

The 29 DEG/ENaCs of *N. vectensis* are distributed in almost all subclades, except for subclades IV and VI, which contain only DEG/ENaCs of medusozoans but not anthozoans. The closest relatives of ASICs and BASIC are NeNaC23 and NeNaC2; we found that NeNaC2 also senses protons but not NeNaC23 (see below). NeNaC1 and NeNac24 are close relatives of peptide-gated HyNaCs, but we have not identified a peptide directly opening these two NeNaCs (see below). The diversification of NeNaCs in subclades III and V (Fig. 1a) is striking. Because subclade III contains no medusozoan DEG/ENaCs and subclade V only two (one of them is HyNaC12), we speculate that the diversification of anthozoan DEG/ENaCs in these two subclades serves anthozoan-specific functions.

Placing the sponge DEG/ENaCs as a sister clade to all other metazoan DEG/ENaCs inferred a scenario in which DEG/ENaCs of sponges and non-sponge animals split and diverged from an ancestral DEG/ENaC. The next event in the evolution of DEG/ENaCs would have been the early split into clade A and clade B in non-sponge metazoans and the species-specific diversification within these two clades. *Trichoplax*, Cnidaria, and Bilateria all contain representatives of both clades (Fig. 1a), demonstrating that their evolution separated after this split. For ctenophores, the situation is ambiguous. On the one hand, ctenophore DEG/ENaCs are monophyletic. On the other hand, our phylogenetic analysis suggests that they are sister clade to subclades I–III and evolved after the split of subclade IV from subclades I–III. This would imply that Ctenophora had the same set of DEG/ENaCs as Cnidaria but lost their clade B gene(s). Broader sampling of DEG/ENaCs from multiple ctenophore species might help to resolve this ambiguity.

Placing ctenophore DEG/ENaCs as a sister clade to all other metazoan DEG/ENaCs inferred a scenario in which sponges are lost from clade A and their DEG/ENaCs form a sister clade to subclades V–VI of clade B (Supplementary Figs. 3, 5). However, with both taxa as outgroups and using two methods of phylogenetic reconstruction, the trees always recovered the same major clades (A and B) and the same subclades (I, II, III, IV, V, and VI). Thus, in both scenarios, it appears that at the base of bilaterians, cnidarians, and placozoans (*Trichoplax*) there have been only two DEG/ENaCs, one gave rise to clade A and the other to clade B, revealing extensive lineage-specific diversifications of DEG/ENaCs from a small set of ancestral channels.

The position of ASICs in clade A and mammalian ENaCs in clade B confirms the deep phylogenetic separation of the two mammalian DEG/ENaC subfamilies, excluding that ENaC evolved from the ASIC lineage as was suggested previously[11]. It is remarkable that the stimuli, to which DEG/ENaCs respond, are either omnipresent, like protons, divalent cations, salt, and mechanical stimuli or are considered to be very ancient ligands, like neuropeptides[49]. Collectively, the lineage-specific diversifications of DEG/ENaCs suggest that these versatile channels were repeatedly co-opted for the detection of different stimuli in different animals, contributing to their evolution and diversification.

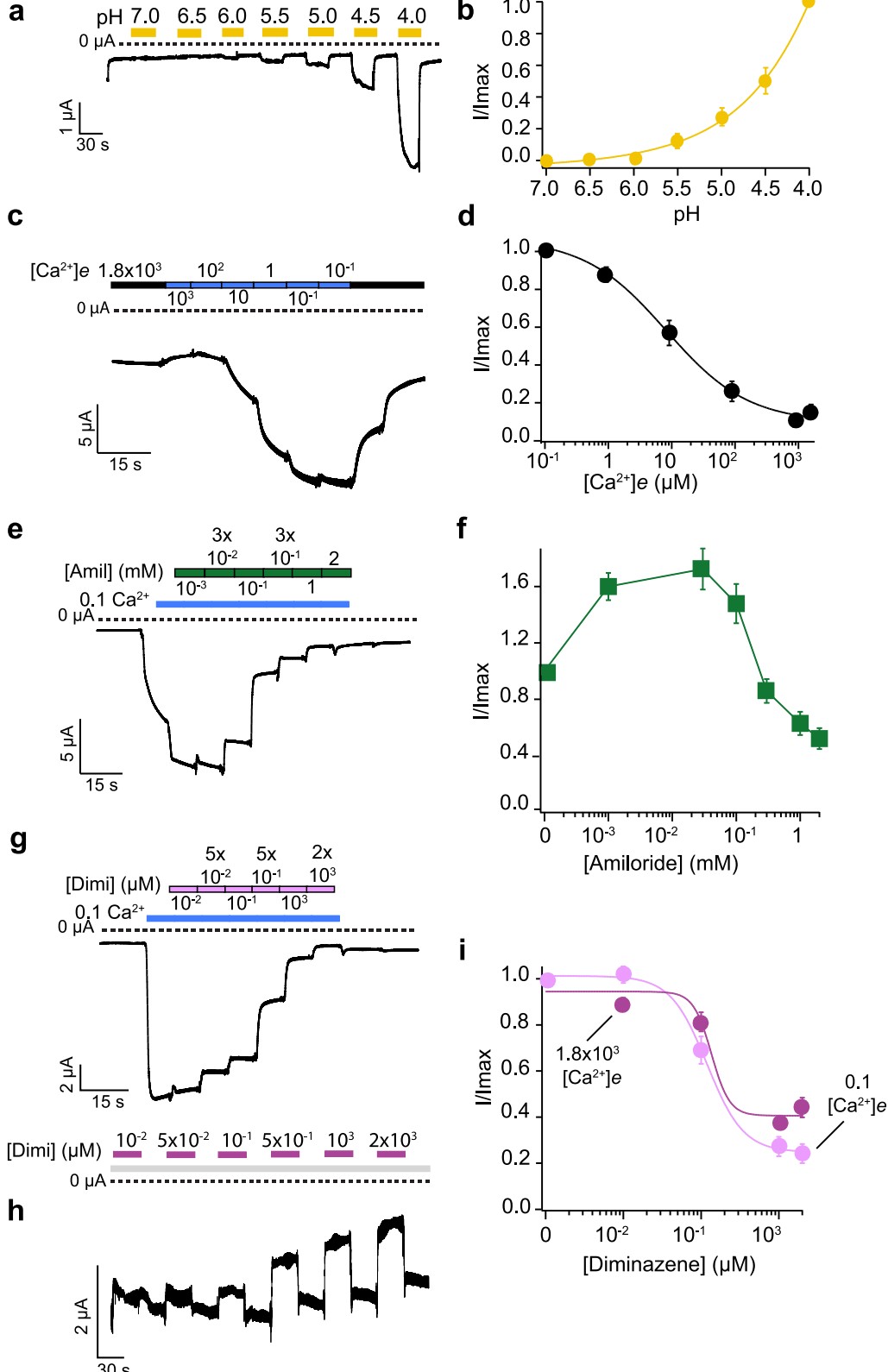

**Fig. 4 NeNaC14 is sensitive to high proton concentrations and NeNaC8 is inhibited by extracellular Ca²⁺. a** Representative current trace of NeNaC14 being activated by increasingly more acidic pH levels (yellow bars). **b** pH–response curve. **c** Representative current trace showing opening of NeNaC8 by lowering $[Ca^{2+}]_e$. $[Ca^{2+}]_e$ is indicated in μM. **d** Concentration-response curve. $n = 15$. **e** Representative current trace showing inhibition of NeNaC8 by amiloride; NeNaC8 was openend by reducing $[Ca^{2+}]_e$ to 100 nM (blue line). **f** Concentration–response curve. $n = 11$. **g** Representative current traces showing inhibition of NeNaC8 by diminazene, dissolved in bath solution containing 100 nM $Ca^{2+}$ (light pink bars). **h** Representative current traces showing inhibition of NeNaC8 by diminazene, dissolved in bath solution containing 1.8 mM $Ca^{2+}$ (dark pink bars). **i** Concentration–response curves. $n = 14$. *Error bars* represent S.E.M.; Concentration–response curves in **b**, **d**, and **i** were fitted with the *Hill* equation.

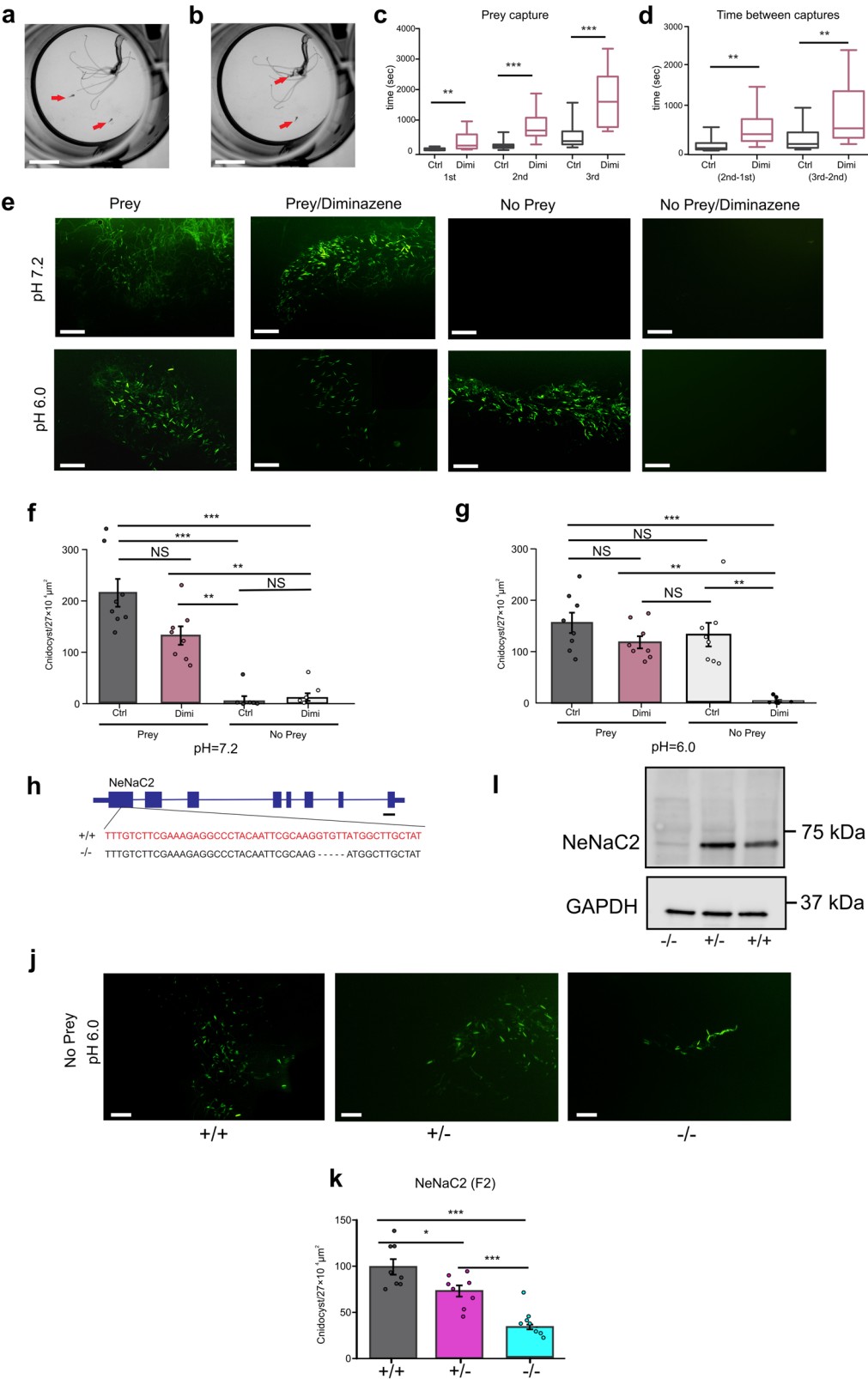

We found that NeNaC2 is a proton-sensitive channel expressed in cnidocytes. The fact that the expression patterns of NeNaC2 according to ISH (Fig. 2) and scRNA-seq data[35] is not overlapping with other NeNaCs and that it is efficiently expressed in *Xenopus* oocytes as a homomer (Fig. 3a), strongly suggests that NeNaC2 is also expressed as a homomer in cnidocytes. Previous studies have indicated that acidic pH induces cnidocyst discharge in *N. vectensis*[46], and we found that NeNaC2 has a role in this mechanism. Pharmacological blockade by diminazene abolished acid-induced cnidocyte discharge (Fig. 5f) and genetic knockout of NeNaC2 strongly reduced it (Fig. 5j, k). Further studies are required to understand the proton sources in *N. vectensis*, and the molecular components and ecological implications for the cnidocyst discharge at acidic pH.

**Fig. 5 Diminazene inhibits the time of capturing *Artemia* nauplii at pH 7.2 and cnidocyst discharge at acidic pH. a** One 3-month-old anemone in a well, ready to capture two swimming *Artemia* nauplii (red arrows). **b** the *Nematostella* polyp captured one of the nauplii with one of its tentacles and drags it to its pharynx. **c** Box plot with whiskers showing the time in seconds in which polyps capture the first, second, and third nauplius in the absence and in the presence of diminazene (100 µM). **d** Box plot with whiskers showing the differences in time between the capture of the second and the first, and the third and the second nauplius in the absence and in the presence of diminazene (100 µM). **e** Pictures of discharged cnidocysts of *NvNcol3::mOrange2* positive organisms at two pH values (7.2 and 6.0), with and without prey extract and with and without diminazene. **f** Bar graph showing the number of discharged cnidocysts (mean ± S.E.) at pH 7.2 with and without prey extract and with and without diminazene. **g** Bar graph showing the number of discharged cnidocysts (mean ± S.E.) at pH 6.0 with and without prey extract and with and without diminazene. **h** Gene model of NeNaC2 showing exons and introns; DNA sequences at the bottom show the sequence of the WT and the mutant; the mutant shows a deletion of five nucleotides. The gene region that the sgRNA targeted is indicated in red. **i** Western blot from NeNaC2 F2 individuals of the three genetic pools showing that the protein was not expressed in NeNaC2$^{(-/-)}$ animals; GAPDH was used as a control. **j** Pictures of discharged cnidocysts of the F2 progeny at pH 6.0 and without prey extract from the three genetic pools that were *NvNcol3::mOrange2* positive. **k** Bar graph showing the number of discharged cnidocysts (mean ± S.E.) of the F2 progeny from NeNaC2 of the three genetic pools at pH 6.0 and without prey extract. Unpaired student's *t* test from 15 individuals in each treatment for data of **c** and **d**. One-way ANOVA of eight organisms per treatment with Bonferroni post-hoc multiple comparisons test for data of **f** and **g**. One-way ANOVA from eight individuals per treatment with Tukey post-hoc multiple comparisons test for **k**. NS, no significant differences, *, ≤0.01; **, ≤0.001, **, ≤0.0001. Scale bars: **a**, **b** 1 mm; **e** 100 µm; **h** 100 bp; **j** 100 µm.

Different ion channels (i.e., Ca$_v$, TRP, K$^+$ channels) are expressed in cnidocytes and are involved in cnidocyst discharge[50]. These channels are triggered by chemo- and/or mechano- sensitive cues (e.g., prey extract), indicating that there is mechanistic redundancy in controlling discharge. Some of these channels such as the voltage-gated Ca$^{2+}$ channels seem to be conserved in different cnidarian species[51–53], while others might be lineage-specific innovations[54]. In *H. magnipapillata,* a cyclic nucleotide-gated channel in non-cnidocyte neurons regulates cnidocyst discharge by light[54]. Thus, cnidocytes integrate mechanical and chemical cues into their discharge behavior. Our results show that protons are one of these environmental cues and that NeNaC2 is the sensor of it.

NeNaC2 is a member of a monophyletic subclade containing ASICs from deuterostomes, TadNaCs, BASICS, and other paralog copies of Anthozoa (but not Medusozoa) and the two degenerins Del-9 and Del-10. While ASICs are proton-sensitive[7], BASIC is not[9]. Among TadNaCs, only TadNaC2 and TadNaC6 have been investigated in detail. TadNaC2 is also activated by protons with low sensitivity (pH$_{50}$ = 5.1)[29] and TadNaC6 is a constitutively open channel which is modulated by alkaline pH[15]. While Del-9 is also activated by protons with low sensitivity (pH$_{50}$ = 4.3)[55], the activating stimulus of Del-10 is currently unknown. It is therefore tempting to conclude that the last common ancestor of Cnidaria, Placozoa and deuterostomes had a proton-sensitive DEG/ENaC, which was at the base of this subclade, and that multiple duplication events occurred during the later evolution of these lineages, including independent gene losses in some protostomes and medusozoans. In this scenario, the gating mechanisms of BASIC and TadNaC6 would be derived or independent features. However, the molecular determinants for proton activation of TadNaC2 have been investigated in some detail and the authors found that they are fundamentally different from those of deuterostome ASICs[29]. The lack of the critical histidine residue at the beginning of β1 in NeNaC2, although not conclusive, is a first indication that the molecular determinants of proton activation of NeNaC2 are also different from canonical ASICs. Together, these findings suggest that the proton-sensitivity of ASICs, TadNaC2, Del-9, and NeNaC2 evolved independently from a proton-insensitive DEG/ENaC at the base of this subclade.

Recently, it has been reported that the *C. elegans* DEG/ENaCs ACD-5 and FLR-1 sense protons in the intestinal epithelium of *C. elegans*. ACD-5 shows maximal current at around pH 6.0 and is inhibited by both lower and higher pH; the FLR-1-containing channel is inhibited by acid[56], resembling TadNaC6[15]. ACD-1, which is expressed in *C. elegans* glia, is also inhibited by acid[57]. ACD-1, ACD-5, and FLR-1 are closely related to each other and belong to a monophyletic clade that is sister to subclade VI (Fig. 1a, b and Supplementary PDF 1). *C. elegans* ACD-2 is a close

relative of ACD-1, ACD-5, and FLR-1 (Fig. 1) and has recently been reported to be activated by protons with low sensitivity (pH$_{50}$ = 5.0)[55]. Belonging to the same clade is *Drosophila* PPK1, which is expressed in multidendritic sensory neurons, and has also been reported to sense acid[58]. These channels are in relatively distant subclades from ASICs and NeNaCs within clade A (Fig. 1a). Moreover, the gating mechanism of some of these channels seems to be different (activation vs inhibition by protons), further suggesting that proton sensitivity evolved independently multiple times in this ion channel superfamily. A definite answer to the question whether proton-gating of some distant family members reflects an ancient feature of DEG/ENaCs awaits the identification of the molecular determinants of proton activation in several distant family members.

Previous studies characterized HyNaCs, which form heterotrimers gated by two *Hydra* RFamide neuropeptides[14,26]. Peptide-gated HyNaCs all cluster in subclade II (Fig. 1a), which contains only cnidarian DEG/ENaCs. Because, according to current knowledge, none of the DEG/ENaCs in the most closely related subclade I are gated by peptides[59], it is likely that peptide-gating evolved in cnidarian subclade II members. This implies that this feature evolved independently in the peptide-gated clade B members (FaNaCs and MGIC). The ligand binding sites of HyNaCs, FaNaCs, and MGIC are currently unknown. Their identification will help to further clarify the independent evolution of activation by peptides in these far-related DEG/ENaCs.

NeNaC1 and NeNaC24 are monophyletic with HyNaCs, making them good candidates for peptide-gated channels. Surprisingly, however, despite screening many different *Nematostella* neuropeptides, we were not able to identify a peptide that activated NeNaC1 or NeNaC24 (either expressed alone or in combination). However, the possibility remains that a currently unknown peptide activates these two NeNaCs.

Our study reveals the phylogenetic complexity of the DEG/ENaC gene family. Moreover, it shows that proton sensitivity of DEG/ENaCs has evolved independently and repeatedly in different lineages. For many DEG/ENaCs, including almost all of the NeNaCs, function and activating stimuli remain to be discovered. This functional heterogeneity is in stark contrast to the relatively high functional conservation of other ion channel families, such as voltage-gated ion channels, in cnidarians and bilaterians[60–62]. Our analysis of NeNaC2 revealed an unanticipated role for a proton-sensitive DEG/ENaC in cnidocysts discharge. However, the physiological relevance of this mechanism as well as the source of protons under native conditions during predation should be further investigated. Generally, we expect that further study of DEG/ENaCs will reveal more surprising roles for these versatile channels in diverse animals.

## Methods

**Sea anemone culture**. *Nematostella* planulae, metamorphic phase individuals, juveniles, and mature polyps were grown in *Nematostella* medium, composed of artificial seawater (Red Sea) at 16‰ in the dark at 22 °C. Anemones were fed with *Artemia salina* nauplii three times a week and the induction of gamete spawning for the mature polyps was performed as previously described[63].

**Gene identification and molecular phylogenetic analyses**. ASICs and ENaCs sequences from humans were used as a query and blasted again the genome and transcriptomics datasets of *Nematostella vectensis*[64] (Supplementary Table 4) and in the genomes of selected species from the phylum Cnidaria (Subphylum Anthozoa and Medusozoa): *Scolanthus callimorphus*[65], *Pocillopora damicornis*[66], *Actinia tenebrosa*[67], *Amplexidiscus fenestrafer*[68], *Stylophora pistillata*[69], *Aurelia aurita*[31], and *Clytia hemisphaerica*[70] (Supplementary Data 1). Additional DEG/ENaC sequences from representatives of different phyla were downloaded from Genbank: *Hydra magnipapillata* (Cnidaria), *Amphimedon queenslandica* (Porifera), *Trichoplax adhaerens* (Placozoa), *Caenorhabditis elegans* (Nematoda), *Drosophila melanogaster* (Arthropoda), *Platynereis dumerilii* (Annelida), *Cornu aspersum* (Mollusca), *Aplysia kurodai* (Mollusca), *Lymnaea stagnalis* (Mollusca), *Aplysia californica* (Mollusca), *Danio renio* (Chordata), and *Homo sapiens* (Chordata). The sequences of *Acanthaster planci* (Echinodermata), *Strongylocentrotus purpuratus* (Echinodermata), *Ptichodera flava* (Hemichordata), and *Branchiostoma belcherii* (Chordata) were obtained from Ref.[12]. The sequences of *Mnemiopsis leidyi* (Ctenophora) were obtained from dataset of[15] (Supplementary Data 1). Amino acid sequences were aligned using MUSCLE which is included in the SEAVIEW software[71,72]. A conservative alignment strategy was employed where all the positions that were spuriously aligned were excluded (by deleting questionable sites and saving the regions of the "reduced" alignment). The first alignment contained 310 sequences, but sequences having many gaps and those that were not informative were deleted for the final alignment (23) (Supplementary Table 5). The final alignment contained 287 sequences with 524 amino acid sites (Supplementary Data 2). The maximum-likelihood phylogenetic trees were constructed using the IQ-Tree Software[73]. WAG + FO + I + G4 was the best fit evolutionary model and −253005.064 was the optimal log-likelihood. Support values of the ML tree were calculated by three different methods: 1000 ultrafast bootstrap replicates[74], 1000 replicates of the Shimodaira–Hasegawa approximate likelihood ratio test (SH-aLRT), and an approximate eBayes test[75]. We also analyzed with Bayesian inference the final alignment applying the model GTR in PhyloBayes MPI v1.8[76]. Two chains were conducted, each with sampling frequency as 1 and constant sites removed (-dc). The convergence of the two chains was checked with tracecomp (maxdiff = 0.1). The consensus tree was obtained using bpcomp from trees sampled of the two chains to compute the posterior probabilities of clades. 10% of the sampled trees were discarded as burn-in. Choosing the best-fitting model was done based on posterior predictive resampling using the amino acid diversity as summary statistic. Four models were tested, LG, GTR, CAT, and CAT-GTR. The molecular trees were visualized by using the FigTree software[77]. Some of the molecular clades were collapsed and the asterisks above and below branches in Fig. 1a represent a support value >65% for the three different methods calculated in IQTREE and Bayesian posterior probabilities >0.65 calculated in PhyloBayes, respectively (Supplementary Figs. 3, 6, 7).

**In situ hybridization (ISH)**. RNA in situ probes of 20 out of 29 NeNaCs were generated. Forward and reverse primers were designed, and the amplicon of each NeNaC probe was amplified using the Advantage 2.0 Polymerase (Takara Bio, Japan) from cDNA of different life stages of *N. vectensis* (900–1500 bp length) (Supplementary Table 6). The amplicons were cloned into the pGEMT plasmid vector (Promega, USA) and then purified using the Quick Plasmid Purification Kit (Thermo Fisher Scientific, USA). Antisense probes were synthesized by in vitro transcription (MEGAScript Kit; Thermo Fisher Scientific) driven by T7 RNA polymerase with DIG incorporation (Roche, Switzerland). ISH was performed as previously described[63].

**Cloning of selected NeNaCs and cRNA synthesis**. Of the 29 discovered NeNaCs, 19 were synthesized and cloned into pRSSP, a custom-made *Xenopus laevis* oocyte vector, for expression in oocytes[78]. The other 10 NeNaCs were expressed at very low levels in just a handful of cells and some of them had incomplete gene models (mostly missing their ends). They were, therefore, not included in our functional analysis. Most of the NeNaCs were directly cloned from cDNAs; a few others were synthesized and cloned via Gene Universal (China). These cDNAs were then turned into capped cRNA, using the mMessage mMachine kit (Thermo Fisher Scientific) and SP6 RNA polymerase for linearized plasmids.

**Electrophysiology**. Stage V–VI oocytes were collected from anesthetized *Xenopus laevis* adult females (2.5 g/l tricainemethanesulfonate for 20–30 min). Anesthetized frogs were killed after the final oocyte harvest by decapitation. Animal care and experiments followed approved institutional guidelines at RWTH Aachen University.

About 8 ng of cRNA was injected into oocytes. Oocytes were kept in OR-2 (Oocyte Ringer solution 2; in mM): 82.5 NaCl, 2.5 KCl, 1 Na$_2$HPO$_4$, 1 MgCl$_2$, 1 CaCl$_2$, 5 HEPES, 0.5 g/l PVP, pH 7.3) and incubated at 19 °C for 24–72 h.

We used TEVC for recording of whole-cell currents. Currents were recorded with a TurboTec 03X amplifier (npi electronic, Tamm, Germany) using an automated, pump-driven solution exchange system together with the oocyte-testing carrousel controlled by the interface OTC-20 (npi electronic). We controlled data acquisition and solution exchange with both CellWorks version 5.1.1 and version 6.2.2 (npi electronic). If not indicated differently, data were acquired at −70 mV holding potential. Data were filtered at 20 Hz and acquired at 1 kHz. Experiments were performed at room temperature. Bath solution contained (in mM) 140 NaCl, 1.8 CaCl$_2$, 1.0 MgCl$_2$, 10 HEPES, pH 7.4. Acidic pH solutions were prepared the same way, except for 10 mM MES as buffer instead of HEPES (pH ranging from 6.5 to 4.0). To avoid activation of the endogenous CaCC of oocytes, some oocytes expressing NeNaC2 were additionally injected with 42 nl EGTA (20 mM, pH 7.4).

Data were collected from oocytes of at least two different animals, if not stated otherwise. Data were analyzed and visualized with the software IgorPro (WaveMetrics, USA), Microsoft Excel 2019 (Microsoft), and Graphpad Prism 6 (Graphpad, USA). Data are presented as mean ± S.D. in the text and as mean ± S.E.M. on the figures, if not stated otherwise.

To determine ion selectivity, we ran voltage ramps from −70 mV to +60 mV in 6 s and calculated ion permeability ratio for monovalent cations $P_{Na}/P_K$ from the shift in $E_{rev}$ when by exchanging Na$^+$ for K$^+$ as the main ion in the extracellular solution, according to the following equation derived from the Goldman–Hodgkin–Katz equation (Equation 1):

$$\frac{P_{Na}}{P_K} = \frac{[K^+]_o}{[Na^+]_o} e^{\left(\frac{\Delta E_{rev}*F}{R*T}\right)}$$

where $\Delta E_{rev} = E_{Na} - E_K$. $E_{Na}$, and $E_K$ were determined using a linear fit between the two values in which the current reversed its sign. $F$ is the Faraday constant, $R$ is the gas constant, and $T$ is temperature in kelvin. To determine $P_{Ca}/P_{Na}$, we used the following equation (Equation (2)):

$$\frac{P_{Ca}}{P_{Na}} = \frac{[Na^+]_o(1 + e^{\left(\frac{E_{Ca}*F}{R*T}\right)})}{4[Ca^{2+}]_o e^{\frac{\Delta E_{rev}*F}{R*T}}}$$

where $\Delta E_{rev} = E_{Na} - E_{Ca^{2+}}$. $E_{Na}$ and $E_{Ca2+}$ were calculated same as $E_{Na}$ and $E_K$ above. $F$, $R$, and $T$ have the same meaning as above. The solution used to determine $E_{Ca}$ contained (in mM): 126.5 NMDG-Cl, 10 HEPES, and 10 CaCl$_2$, adjusted to pH 7.4. The solution used to determine $E_{Na}$ contained (in mM): 140 NaCl, 10 HEPES, and 1 CaCl$_2$, adjusted to pH 7.4; the small amount of Ca$^{2+}$ in this solution was considered negligible. *I–V* relationships were corrected for background conductances by subtracting the current measured with voltage ramps at pH 7.4.

Activity of ions was used in all terms of [$c$]. Activity coefficients $f_i$ of single ions $i$ of valence $z$ were calculated using the Davies equation (Equation 3).

$$\log_{10} f_i = -0.509 z_1 z_2 \left(\frac{\sqrt{I}}{1 + \sqrt{I}} - 0.2I\right)$$

where $I$ is the ionic strength, defined as (Equation 4):

$$I = 0.5\sum c_i z_i^2$$

**Prey capture and cnidocyte discharge experiments**. A prey capture experiment was designed to estimate the time that it takes for starving anemones to capture three swimming artemias. Three months old anemones were selected over younger or older ages, because it is easy to observe the capture of the *Artemia* nauplii (capturing one nauplius with one of their tentacles and dragging it to their pharynx for feeding) using a stereomicroscope. Individuals starved for 3–5 d were placed in 24-well plates filled with 1.5 ml of *Nematostella* medium (pH = 7.2) at 18 °C. 45 min after placing the animals into the 24-well plate, each well was checked and only polyps that had their tentacles exposed were selected to carry out the experiment. Three live nauplii were put into each well with a glass pipette and the time that it takes for the polyp to capture them with one of its tentacles and drag it to its pharynx (the 1st, 2nd, and 3rd artemia) was estimated using a stopwatch under an SMZ18 stereomicroscope (Nikon, Japan). The time (s) that it takes for the anemones to capture three nauplii and the time between the captures (2nd–1st and 3rd–2nd) were plotted. Several aspects were considered to carry out this experiment: t0 started when the three nauplii were put into each well and they were capable of swimming; if one of them after placing into the well did not swim, was inactive or sank to the bottom, the experiment was annulled; if one of them attached to the body wall of the anemone, the experiment was annulled; if the anemone contracted or hided its tentacles after placing three nauplii into the well or during the prey capture, the experiment was annulled; the time considered for capturing was when the nauplius was dragged to the pharynx after being captured with one of the tentacles. The prey capture experiment was carried out at normal conditions (pH = 7.2) as a control and in the presence of diminazene (pH = 7.2; 100 µM, Sigma-Aldrich, USA). The nauplii swam normally in the presence of diminazene under these conditions.

For the cnidocyte discharge experiment, the protocol of ref. [50] was followed with some slight modifications: Microscope slides were coated with 25% gelatin from skin of bovine (Sigma-Aldrich). The slides were dried o/n, and the prey extract was prepared as in ref. [50]. Individuals starved for 3–5 d that were older than four months and *NvNCOl3:mOrange2* positive were selected to carry out this experiment. Discharged cnidocysts were identified under a fluorescent Eclipse NiU microscope (Nikon) by using this transgenic reporter line[79]. The anemones were placed in petri dishes with *Nematostella* medium, with or without diminazene (100 μM) and at three different pH concentrations (7.2, 6.0, and 5.5). The pH adjustment of the *Nematostella* medium (before placing the anemones to the petri dishes) was accomplished by adding either HCl (1 M) or NaOH (3 M). The dried gelatin-coated microscope slides were dipped in the prey extract and then presented to the tentacles of the anemones for 5 s. Additional slides were also presented to the anemones without the prey extract. A cover slide was put onto the microscope slide and observed under 20× magnification of the Eclipe NiU microscope. Several pictures were taken for each replicate of the different samples and controls by using a DS-Ri2 camera mounted on the microscope and controlled by an Elements BR software (Nikon). The quantification of the discharged cnidocyst capsules was performed manually by using a tally counter in each of the pictures taken. The area of the picture containing the discharged cnidocysts was estimated and is presented as μm$^2$.

**Generation and characterization of transgenic knockout null mutant line of NeNaC2**. CRISPR/Cas9 genome editing in *Nematostella* embryos was carried out using an established protocol[80]. gRNAs were designed using the online web interface CRISPOR[81]. One sgRNA targeting the first exon NeNaC2 was synthesized in vitro using the MEGAshortscript T7 kit (Thermo Fisher Scientific) using the following oligos for NeNaC2: 5′ TAGGCTACAATTCGCAAGGTGTTA and 5′ AAACTAA-CACCTTGCGAATTGTAG. The reaction mixture was incubated at room temperature for 10 min prior to injection into fertilized eggs. It includes the Recombinant Cas9 protein with NLS sequence (1500 ng /μl⁻¹; PNA Bio, CP0120) and the sgRNA (750 ng/ μl⁻¹). Genomic DNA was extracted in F0 primary polyp individuals by washing the samples three times with MeOH 100% using PCR tubes. The samples were dried at 50 °C for 10 min and then a DNA extraction solution (10 mM Tris pH 8; 1 mM EDTA pH 8; 25 mm NaCl and 200 μg/ml of Proteinase K (Thermo Fisher Scientific) was added to the tubes and incubated for 2 h at 50 °C and then for 5 min at 96 °C (to inactivate the Proteinase K). Mutation analysis was carried out by using the High-Resolution Melting Curve Method (HRM) in the Magnetic Induction Cycler (MIC) qPCR (Bio Molecular Systems, Australia), for the detection of indels in the samples tested. These samples were subsequently used to amplify a PCR fragment, adjacent to the DNA region containing the mutation. Primers for HRM in NeNaC2 are: 5′ ACCCTGACACTACACGGCTTTCGGTTTG (forward) and 5′ TCAGC0ATCCC-TACGGCGAACAGCAGAAT (reverse) 100 bp. Primers for PCR in NeNaC2 are: 5′ TACACACATCTTGGACGCTGCAATAACT (forward) and 5′ TGTGACTGCGG-GAAACTGGATCTCTTCC (reverse) 415 bp. F0 injected animals were raised until they reached their sexual maturity (>4 months old) and then were induced to spawn gametes according to an established protocol[62]. They were crossed with wild-type organisms to obtain an F1 heterozygous progeny. Genomic DNA was extracted from ten individuals of the crosses of each of the tentative F0 founders. Mutation detection was carried out as with the F0 injected animals and F1 progeny carrying mutations was raised until the polyps reached sexual maturity (>4 months old). One F1 heterozygous individual was crossed with other F1 heterozygous individual having the same DNA mutation. F2 progeny from this cross were raised until they were 2 months old. These F2 individuals were placed in either 12 or 24 well plates and individually genotyped (to identify wild type, heterozygous and null homozygous). A piece of the tentacle was cut with a scalpel and the genomic DNA extraction was carried out as with F0 injected animals but instead of doing three washes with MeOH 100%, five were done.

For the generation of the knock-out null homozygous mutants of NeNaC2 bearing the *NvNcol3::mOrange2* reporter, one F0 NeNaC2 mutant was crossed with one polyp carrying the *NvNcol3::mOrange2* transgene. F1 organisms that were *NvNcol3::mOrange2* positive and NeNaC2 heterozygous were identified and raised until they reached sexual maturity. One F1 heterozygous NeNaC2 mutant with *NvNCOl3::mOrange2* background was crossed with another individual F1 heterozygous NeNaC2 having the same mutation. F2 progeny of this cross that was mOrange2 positive were selected and individually genotyped. The prey capture experiment (as above) was carried out for the three genetic pools of the F2 progeny (3 months old) of NeNaC2. The cnidocyte discharge experiment was carried out for the three genetic pools of the F2 progeny (3 months old) of NeNaC2 that were *NvNCol3::mOrange2* positive.

**Western blotting**. For NeNaC2 Western blot, we used the method as previously described in ref. [82]. In brief, custom polyclonal antibodies raised against recombinant fragment antigens generated by immunization of rabbits (GenScript, USA). The sequence of the recombinant fragment was ITGCLSLYDLKLIAAVMSCP-VAQRHFETEEDKKDEDEDDRAEDPVDENPDDTITVSQMWQDFLHTLTLH GFRFVFERGPTHHHHHH. Equal amounts of protein were run on 4–15% Mini-PROTEAN® TGX™ Precast Protein Gel (Bio-Rad, USA) followed by blotting to a Polyvinylidene fluoride (PVDF) membrane (Bio-Rad). Membrane was incubated overnight with polyclonal antibody against NeNaC2 or monoclonal mouse anti-GAPDH (Abcam, UK) with a dilution of 1:1000 at 4 °C overnight, then washed and

incubated for 1 h with peroxidase-conjugated anti-mouse or anti-rabbit antibody (Jackson ImmunoResearch, USA) with a dilution of 1:10,000. Detection was performed with the Clarity™ Max ECL kit (Bio-Rad) according to the manufacturer's instructions and visualized with a CCD camera of the Odyssey Fc imaging system (Li-COR Biociences, USA).

**Statistics and reproducibility**. Data were examined with the Prism software (Graphpad, USA) and considered significant if $p < 0.05$, using unpaired two-tailed Student's *t* test or one-way ANOVA. All experiments were performed with biological replicates to ensure reproducibility as described for each of them separately. The individual numerical measurements for each of them are available in Supplementary Data 3.

**Reporting summary**. Further information on research design is available in the Nature Portfolio Reporting Summary linked to this article.

## Data availability

All data are available in the main text, supplementary materials, and supplementary datasets. Original uncropped blots for Fig. 5i appear in Supplementary Figs. 13 and 14.

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

## Acknowledgements

We thank S. Joussen for expert technical assistance. This project was funded by the German Israeli Foundation for Scientific Research & Development (GIF) grant I-1449-207.9/2018 to S.G. and Y.M.

## Author contributions

Conceptualization: S.G., Y.M. Methodology: J.M.A.C., A.J.B. Visualization: J.M.A.C., K.F. Investigation: J.M.A.C., K.F., A.J.B., R.A. Supervision: S.G., Y.M. Writing—original draft: J.M.A.C., S.G., Y.M. Writing—review and editing: J.M.A.C., K.F., A.J.B., R.A., S.G., Y.M.

## Funding

## Competing interests

The authors declare that they have no competing interests.
