## [Peer Review File · Communications Biology]

Reviewers' comments:

Reviewer #1 (Remarks to the Author):

The study is great: hypotheses are well tested, multiple approaches were used, and the data are very clean. The only issue I have with the manuscript is the narrative. I understand the emphasis on ENaC evolution but am left wondering about the interesting observation that these channels impact stinging. The authors could at least comment further on the following questions:

- Why do anemones sting in response to acid?
- Why would they do this in the absence of prey?
- How are proton-activated currents transduced to mediate discharge?
- Why would external calcium affect stinging based on the animal's ecology?

Overall, the authors could further discuss function and what motivated their experiments in the context of understanding anemone behavior.

Reviewer #2 (Remarks to the Author):

"Functional analysis in a model sea anemone reveals phylogenetic complexity and a role in cnidocyte discharge of DEG/ENaC ion channels" by Aguilar-Camacho...and Moran is a manuscript reporting the discovery and the biophysical and biological characterization of a subfamily of ion channels from sea anemone. It reports novel, exciting results, and it describes methods transparently and is solid science. The work is very informative on the evolution of channels of the DEG/ENaC family and raises new questions that will inspire further study. I think the paper does a uniquely clear/thorough job of describing evolution within this channel superfamily, compared to other papers in the field. In my opinion, the manuscript deserves publishing in Communications Biology after some small changes that address small matters of accuracy or clarity, detailed below.

The introduction to various DEG/ENaCs is useful. Two small notes on this

- FaNaCs are found in several lophotrochozoans, not "just snails" (second page of introduction) (Dandamudi, J Biol Chem, 2022).
- I don't see evidence that the feature of "all DEG/ENaCs of Hydra are activated by peptides" is shared in some protostomes (second page of introduction). Does this refer to lophotrochozoans? FaNaC/MGIC are not the only lophotrochozoan genes characterized, as suggested by your Fig 1 and the protostome ASICs referred to earlier in the introduction. It clearly doesn't refer to ecdysozoans, where (as described by authors) functions/ligands are diverse.

The description of the major phylogenetic tree is clear and the relationships in the tree make sense. The description in the Methods is also thorough and transparent, but there should be a sentence early in the Results (not just in the Discussion) explaining that trees differ from model to model and from paper to paper, and relationships may be refined in future, e.g. with sequences of more diverse sponges, placozoans, and cnidarians. Without a sentence like this, physiologists and biophysicists will treat the tree as indisputable fact and make spurious conclusions in future.

It is not clear why not all stimuli were tested at all NeNaCs, especially when the experiment is as simple as applying low pH solution to the oocytes.

On the third page of Results, we read that the DEG position is "just proximal to TMD2". It is actually in TMD2.

Supplementary table 1 – Please make the columns in the upper and lower parts of the table

match/line up. Otherwise it is difficult to read/looks like different channels are active on WT or DEG background.

Third page of results – change mA to microA?

Fourth page of Results – We read that (1) from chelator experiments, currents are not affected by removal of Ca²⁺/activation of CaCC, and NeNaC2 is not Ca²⁺ permeable and (2) from extracellular cation substitution and reversal potential experiments that Ca²⁺ is almost as permeable as Na⁺. Certainly in Fig 3C (reversal potential experiments) it looks like the Ca²⁺ permeability is much lower than Na⁺.

I'm not sure that 1 mM is a "low apparent IC₅₀" for amiloride. It is a higher IC₅₀ than at many other DEG/ENaC channels.

"In the presence of prey extract, cnidocysts were similarly discharged at pH 7.2 and at acidic pH (pH 6.0 or 5.5) (Fig. 5e-g and Supplementary Fig 4a, b). Moreover, diminazene did not affect the cnidocyst discharge induced by prey extract at pH 7.2 and pH 6.0," I might be missing something, but looking at the images and data in Fig 5e-g, all of the interpretations in this text seem wrong to me: in the presence of prey, pH 6.0 seems to elicit much more cnidocyte discharge than pH 7.2 in Fig 5e; conversely, pH 7.2 seems to elicit more cnidocyte discharge than pH 6.0 in Fig 5f,g; and dminizene seems to decrease discharge at pH 7.2 (Fig 5e and Fig 5f).

Discussion – "a proton-sensitive DEG/ENaC in *N. vectensis* that is not directly related to ASICs" – If I interpret this correctly, I think this is helpful discussion of DEG/ENaC/ASIC diversity. But perhaps the authors could better explain if/that they are referring to ASICs as a specific family of proton-gated channels or as proton-gated channels in general.

Discussion/Introduction – I think numerous DEG/ENaC studies suggest that ASICs and ENaCs are distant cousins. Despite the suggestion from ref 12 that ENaCs might have emerged from ASICs, I think it's not accurate to say that the present study "reveals their deep phylogenetic separation, excluding that ENaC evolved from the ASIC lineage as was suggested previously" in the Discussion or that the existing notion is that ENaCs and ASICs are close cousins in the Introduction.

Discussion – I would hardly blame the authors for missing something on proton-gated channels this year, given the burst in the literature, but DEL-9 is activated by protons (Kaulich, Biorxiv, 2022), so the authors might like to refine the relevant sentence in their discussion.

If find the discharge experiments good (despite the presentation/interpretation of data mentioned above). I do wonder about the "time taken to catch prey". I understand this means from (a) adding shrimp through to (b) reaching mouth. But is there a useful and easily distinguishable difference between capture (time between adding shrimp and being under control of tentacle) and pulling (time between being under control of tentacle and reaching mouth), especially if the authors are discussing protons affecting discharge.

Reviewer #3 (Remarks to the Author):

This paper presents a phylogenetic analysis of the DEG/EnaC channel family from across the entire metazoan lineage and begins functional characterization of *Nematotella vectensis* (sea anemone) Degenerins (called NeNacs) via heterologous expression and genetic KO of NeNac2. The authors identify two major clades of Deg/Enacs which they interpret as originating prior to the radiation of Trichoplax, Cnidaria and Bilateria. They suggest absence of channels from either clade in any of these groups is likely to represent gene loss. Their functional analysis of several NeNacs shows proton-

gating (or sensitivity in one case), and they propose proton gating has evolved multiple times in the channel family based on the position of proton-sensitive channels within their phylogeny. Finally, they show that NeNac2 is expressed in cnidocytes and enhances cnidocyte discharge at low pH. It however has no effect on discharge related to prey capture in normal conditions.

The manuscript at times gets caught between trying to be a comprehensive description of the NeNac family and trying to tell substantive stories that could advance the Deg/ENac field. It does a better job of the former, providing a descriptive data set that scientists in the field could use to begin functional genetic analysis of NeNac functions in vivo. The authors have indeed started that process themselves with NeNac2. The broader impact on the field will be far more limited because their conclusions on the evolutionary history of the Deg/Enac family, the evolution of proton gating within the family, and the importance of NeNac2 in cnidocyte discharge are not supported by unambiguous interpretations of the datasets presented. As such, it does not seem suitable for publication in *Communications Biology* in its current form. Specific criticisms and strengths are outlined below.

Comments to authors

1. One of the major issues with the manuscript is selective and overly optimistic interpretation of the phylogeny presented. While the phylogeny is interesting as broader analysis of the gene family than previously conducted, there are a number of caveats to its interpretation that weaken some of the authors claims and need to be addressed. First and foremost, the authors are making strong interpretations of branches with extremely short lengths at the base of the tree. This has often lead to contention for important questions in phylogenetics, as interpretations are based on very little phylogenetically-informative data and often come out differently in different hands with small differences in alignment or sequence selection. They use 3 different statistical tests to support their findings and interpretations, but this is not the right approach because they are all applied to a single tree. They should instead select a 2nd and maybe 3rd tree-building algorithm – robust nodes are recovered with multiple algorithms, while weak nodes may only show up in one. It would help here to know which nodes depend on the author's implementation of Maximum Likelihood, and which might be recovered using different methods such as Bayesian inference, distance methods or just Maximum Likelihood under a different substitution model. Alternatively, the authors should be far more cautious about their interpretation of clades dependent on extremely short root branches. How many sites are even informative to the clade? Is it even double digits? Deg/Enacs are challenging for phylogenetic analysis due to their extreme divergence, and thus require a bit more caution and rigor.

2. Is this an unrooted phylogeny? What drove root selection? This is important because it affects some of the interpretations the authors discuss. For instance, as rooted, the tree suggests Ctenophore channels belong to clade A and they authors interpret clade B has been lost in ctenophores. But what if the tree is instead rooted between ctenophores and the other animals under the assumption ctenophores, not sponges, are the outgroup? Now doesn't this place sponge within clade B and suggest it has lost Clade A? The authors acknowledge Deg losses in freshwater sponge already, so isn't this also a reasonable interpretation? Sponge sequence divergence could just be from reduced selection? The point is that the tree as presented is not the only reasonable interpretation based on the information the authors provide. This should be acknowledged and discussed in its entirety, or the authors could just remove the relevant conclusions they do make. Similarly, its probably not wise to wade into the ctenophore-first vs sponge-first debate with a single highly divergent gene family given the issue is contentious even with extremely large data sets.

3. The authors suggest a high diversity of deg/enac channels in ancestral animals both in their intro and discussion, but is this actually what the phylogeny says? My reading is it does not at all say this. Doesn't it say there is separate lineage-specific diversification from a small set of ancestral channels? Just 2 if clades A and B hold together? And still only two present in multiple lineages of cnidarians or bilaterians? So that most diversification has occurred after divergence of the cnidarian and bilaterian classes? The diversification patterns and timing are very interesting but interpretations are not consistent with the data as presented.

4. The authors pose multiple evolutions for pH sensitivity based on finding pH-sensitive NeNaC channels in clades A and B, but is this the only interpretation? There is very little functional data,

especially with regards to proton gating, across large swaths of the tree. Can the authors rule out that the ancestral Deg/Enac was pH gated? If so, then there might be several isolated losses of proton gating rather than multiple evolutions.

5. The author's genetic/behavioural data showing NeNac2 supports cnidocyte discharge at low pH seems solid, but what is the physiological significance? Seawater has high pH and NeNac2 does not appear to be involved in prey-dependent discharge. Most cnidarians don't have channels closely related to NeNac2, so this seems unlikely to be a fundamental mechanism regulating cnidocyte discharge. Maybe it just needs more explanation or citations? Similarly, is the calcium-inhibition or NeNac8 physiologically relevant given it happens in a submicromolar concentration in an osmoconformer living in seawater? The data looks good, but is this telling us anything relevant about NeNac function in *Nematostella*? The authors should at least discuss whether these described phenomena might or might not have real physiological significance.

6. The conclusion that NeNac2 doesn't significantly conduct calcium appears at odds with it having a calcium permeability almost as high as sodium and far higher than potassium. Is this interpretation based solely on native calcium activated chloride currents and the assumption that EGTA will chelate them effectively? And that protons released from EGTA will not influence results? Some sort of control for EGTA effectiveness in blocking the native current could help (and maybe use Na-EGTA or BAPTA?) Or is their literature to cite that this is an effective method? Regardless, the high calcium permeability suggests the channel contributes to calcium signaling in vivo, no?

Minor points

1. The authors mention that Deg/Enacs are much more diverse on a sequence and functional level compared to voltage-gated channels and imply the sequence divergence is responsible for that functional divergence. Yet what they are really showing here is remarkable conservation of function despite high sequence divergence is it not? This might also be worth mentioning.

2. The color schemes in Figure 1 are unnecessarily complex and make it very hard for the reader to quickly figure out what species individual clades belong to, or to understand which phyla are contributing to a clade. It would work better to use fewer colors to highlight just the key clades, or maybe only use one color/phylum and combine that with text labels for quick deciphering. For example, the text labels are very effective for differentiating HyNacs and NeNacs, but the thin branch colors are hard to figure out.

3. The cloning section in the methods may be a bit too brief. Which 19 were selected? Why weren't the 10 others in the pool? Just lack of data on starts and stops? Or was the selection for some functional reason – like these are the ones they thought would work? Some were synthesized by a specified company, but the rest are not specified? Easy to fix but incongruent and somewhat incomplete as is.

4. Is there a paper to cite that the EGTA injection as done effectively chelates calcium transients in oocytes? Would the potential for intracellular acidification cause issues for Deg/Enac function? I think probably not, but maybe there is a reference to boost confidence?

5. The data comparing NeNac2 H141A to WT is not sufficient for concluding that H141A does or does influence apparent pH sensitivity or play a major role in gating. It could be explored in more detail, removed entirely (probably the best easy choice), or the authors could soften statements and say the mutant has altered gating and this needs further exploration to exactly determine how it is altered. Its too much to say it is dispensible for proton gating when the currents are a small fraction of WT currents unless the authors know that surface expression or conductance are altered in a way that could account for the differences in current size.

6. Some of the figure legends are not complete stand alone descriptions and some minor editing would help the reader.

Response letter:

Reviewer #1 (Remarks to the Author):

The study is great: hypotheses are well tested, multiple approaches were used, and the data are very clean.

Thank you for your positive comments.

The only issue I have with the manuscript is the narrative. I understand the emphasis on ENaC evolution but am left wondering about the interesting observation that these channels impact stinging. The authors could at least comment further on the following questions:

- Why do anemones sting in response to acid?
- Why would they do this in the absence of prey?
- How are proton-activated currents transduced to mediate discharge?
- Why would external calcium affect stinging based on the animal's ecology?

Overall, the authors could further discuss function and what motivated their experiments in the context of understanding anemone behavior.

While we find all the questions raised by the reviewer very interesting, we are afraid that answering them would be quite speculative at this stage. Cnidocytes were shown in different earlier studies to discharge in the lab at acidic conditions (e.g., Karabulut et al. 2022 Nat. Commun. 13: 3494). However, how this is achieved in a natural setting is a completely uncharted area of research and we feel that just guessing or raising speculations at this point will not serve the readers. We mention this now also at the end of the discussion.

Reviewer #2 (Remarks to the Author):

“Functional analysis in a model sea anemone reveals phylogenetic complexity and a role in cnidocyte discharge of DEG/ENaC ion channels” by Aguilar-Camacho...and Moran is a manuscript reporting the discovery and the biophysical and biological characterization of a subfamily of ion channels from sea anemone. It reports novel, exciting results, and it describes methods transparently and is solid science. The work is very informative on the evolution of channels of the DEG/ENaC family and raises new questions that will inspire further study. I think the paper does a uniquely clear/thorough job of describing evolution within this channel superfamily, compared to other papers in the field. In my opinion, the manuscript deserves publishing in Communications Biology after some small changes that address small matters of accuracy or clarity, detailed below.

Thank you for your positive comments.

The introduction to various DEG/ENaCs is useful. Two small notes on this
- FaNaCs are found in several lophotrochozoans, not “just snails” (second page of introduction) (Dandamudi, J Biol Chem, 2022).

Thank you for reminding us of the important paper by Dandamudi and colleagues. We changed the text accordingly.

- I don't see evidence that the feature of "all DEG/ENaCs of Hydra are activated by peptides" is shared in some protostomes (second page of introduction). Does this refer to lophotrochozoans? FaNaC/MGIC are not the only lophotrochozoan genes characterized, as suggested by your Fig 1 and the protostome ASICs referred to earlier in the introduction. It clearly doesn't refer to ecdysozoans, where (as described by authors) functions/ligands are diverse.

We rephrased this sentence to make its meaning clearer.

The description of the major phylogenetic tree is clear and the relationships in the tree make sense. The description in the Methods is also thorough and transparent, but there should be a sentence early in the Results (not just in the Discussion) explaining that trees differ from model to model and from paper to paper, and relationships may be refined in future, e.g. with sequences of more diverse sponges, placozoans, and cnidarians. Without a sentence like this, physiologists and biophysicists will treat the tree as indisputable fact and make spurious conclusions in future.

Following the reviewer's suggestion, a sentence addressing those issues was added to the Results.

It is not clear why not all stimuli were tested at all NeNaCs, especially when the experiment is as simple as applying low pH solution to the oocytes.

From supplementary table 1, which summarizes the stimuli that were applied to individual NeNaCs, it can be seen that a solution with a low concentration of divalent cations was indeed applied to all 15 NeNaCs. Likewise, pH 4.0 was applied to all 15 NeNaCs, except NeNaC10. We did not apply diminazene or amiloride to oocytes expressing some NeNaCs that never displayed obviously increased leak currents or that were not activated by low divalent solution.

On the third page of Results, we read that the DEG position is "just proximal to TMD2". It is actually in TMD2.

Thank you for the comment. We changed the text accordingly.

Supplementary table 1 – Please make the columns in the upper and lower parts of the table match/line up. Otherwise it is difficult to read/looks like different channels are active on WT or DEG background.

Thank you for the comment. We changed the table accordingly.

Third page of results – change mA to microA?

Yes, error fixed.

Fourth page of Results – We read that (1) from chelator experiments, currents are not affected by removal of Ca²⁺/activation of CaCC, and NeNaC2 is not Ca²⁺ permeable and (2) from extracellular cation substitution and reversal potential experiments that Ca²⁺ is almost as permeable as Na⁺. Certainly in Fig 3C (reversal potential experiments) it looks like the Ca²⁺ permeability is much lower than Na⁺.

Thank you for the comment. We indeed made a mistake when calculating relative permeabilities. We recalculated them and found that P_{Na}/P_{Ca} is indeed 0.2, which confirms a relatively low Ca^{2+} permeability of $NeNaC2$.

I'm not sure that 1 mM is a "low apparent IC_{50} " for amiloride. It is a higher IC_{50} than at many other DEG/ENaC channels.

What we really meant was "low apparent affinity". We fixed this error.

"In the presence of prey extract, cnidocysts were similarly discharged at pH 7.2 and at acidic pH (pH 6.0 or 5.5) (Fig. 5e-g and Supplementary Fig 4a, b). Moreover, diminazene did not affect the cnidocyst discharge induced by prey extract at pH 7.2 and pH 6.0," I might be missing something, but looking at the images and data in Fig 5e-g, all of the interpretations in this text seem wrong to me: in the presence of prey, pH 6.0 seems to elicit much more cnidocyte discharge than pH 7.2 in Fig 5e; conversely, pH 7.2 seems to elicit more cnidocyte discharge than pH 6.0 in Fig 5f,g; and diminazene seems to decrease discharge at pH 7.2 (Fig 5e and Fig 5f).

Figure 5e presents images from individual experiments that are indented to illustrate the outcome of the prey capturing experiments. The bar graphs in panels f and g present the summary data. There was indeed a slight reduction in discharge at pH 6.2, which was not significant, however. This was shown in Supplementary Figure 4b (now Fig. S11). Similarly, as indicated in Fig 5 f and g, the decrease in discharge by diminazene was also not statistically significant. To provide additional information, we now describe the respective comparisons in more detail and indicate their p-values in the text.

Discussion – "a proton-sensitive DEG/ENaC in *N. vectensis* that is not directly related to ASICs" – If I interpret this correctly, I think this is helpful discussion of DEG/ENaC/ASIC diversity. But perhaps the authors could better explain if/that they are referring to ASICs as a specific family of proton-gated channels or as proton-gated channels in general.

Thank you for the comment. We changed the text to make clearer that we are referring to the ASIC subfamily of deuterostomes.

Discussion/Introduction – I think numerous DEG/ENaC studies suggest that ASICs and ENaCs are distant cousins. Despite the suggestion from ref 12 that ENaCs might have emerged from ASICs, I think it's not accurate to say that the present study "reveals their deep phylogenetic separation, excluding that ENaC evolved from the ASIC lineage as was suggested previously" in the Discussion or that the existing notion is that ENaCs and ASICs are close cousins in the Introduction.

Thank you for the comment. We agree that previous studies already suggested that ASICs and ENaC are distant cousins. Nevertheless, because recent reviews like ref 12 still propose a scenario in which the two prominent mammalian subfamilies directly evolved from each other (there are also papers proposing that ASICs evolved from the the more "simple" ENaC), we still wanted to underline their deep phylogenetic separation. We rephrased the respective sentences in Introduction and Discussion to make clearer that this is not a novel and major finding of our study.

Discussion – I would hardly blame the authors for missing something on proton-gated channels this year, given the burst in the literature, but DEL-9 is activated by protons

(Kaulich, Biorxiv, 2022), so the authors might like to refine the relevant sentence in their discussion.

Thank you for making us aware of this paper by Kaulich et al. We now cite this paper and refined the respective sentence in the Discussion.

If find the discharge experiments good (despite the presentation/interpretation of data mentioned above). I do wonder about the “time taken to catch prey”. I understand this means from (a) adding shrimp through to (b) reaching mouth. But is there a useful and easily distinguishable difference between capture (time between adding shrimp and being under control of tentacle) and pulling (time between being under control of tentacle and reaching mouth), especially if the authors are discussing protons affecting discharge.

Thank you for this suggestion. However, after checking the available data it seems that we cannot find a distinguishable difference that can be informative.

Reviewer #3 (Remarks to the Author):

This paper presents a phylogenetic analysis of the DEG/EnaC channel family from across the entire metazoan lineage and begins functional characterization of *Nematotella vectensis* (sea anemone) Degenerins (called NeNacs) via heterologous expression and genetic KO of NeNac2. The authors identify two major clades of Deg/Enacs which they interpret as originating prior to the radiation of Trichoplax, Cnidaria and Bilateria. They suggest absence of channels from either clade in any of these groups is likely to represent gene loss. Their functional analysis of several NeNacs shows proton-gating (or sensitivity in one case), and they propose proton gating has evolved multiple times in the channel family based on the position of proton-sensitive channels within their phylogeny. Finally, they show that NeNac2 is expressed in cnidocytes and enhances cnidocyte discharge at low pH. It however has no effect on discharge related to prey capture in normal conditions.

The manuscript at times gets caught between trying to be a comprehensive description of the NeNac family and trying to tell substantive stories that could advance the Deg/ENac field. It does a better job of the former, providing a descriptive data set that scientists in the field could use to begin functional genetic analysis of NeNac functions in vivo. The authors have indeed started that process themselves with NeNac2. The broader impact on the field will be far more limited because their conclusions on the evolutionary history of the Deg/Enac family, the evolution of proton gating within the family, and the importance of NeNac2 in cnidocyte discharge are not supported by unambiguous interpretations of the datasets presented. As such, it does not seem suitable for publication in *Communications Biology* in its current form. Specific criticisms and strengths are outlined below.

Comments to authors

1. One of the major issues with the manuscript is selective and overly optimistic interpretation of the phylogeny presented. While the phylogeny is interesting as broader analysis of the gene family than previously conducted, there are a number of caveats to its interpretation that weaken some of the authors claims and need to be addressed. First and foremost, the authors are making strong interpretations of branches with extremely short lengths at the base of the tree. This has often lead to contention for important questions in phylogenetics, as interpretations are based on very little phylogenetically-informative data and often come out differently in different hands with small differences in alignment or sequence selection. They use 3 different statistical tests to support their findings and interpretations, but this is not the

right approach because they are all applied to a single tree. They should instead select a 2nd and maybe 3rd tree-building algorithm – robust nodes are recovered with multiple algorithms, while weak nodes may only show up in one. It would help here to know which nodes depend on the author’s implementation of Maximum Likelihood, and which might be recovered using different methods such as Bayesian inference, distance methods or just Maximum Likelihood under a different substitution model. Alternatively, the authors should be far more cautious about their interpretation of clades dependent on extremely short root branches. How many sites are even informative to the clade? Is it even double digits? Deg/Enacs are challenging for phylogenetic analysis due to their extreme divergence, and thus require a bit more caution and rigor.

We thank the reviewer for his comment. We now also performed a Bayesian analysis, which recovered a very similar phylogeny as Maximum Likelihood analysis. We indicate in Figure 1 which nodes were recovered with both types of analysis and show the results from Bayesian analysis in Supplementary Figures 2 and 3.

2. Is this an unrooted phylogeny? What drove root selection? This is important because it affects some of the interpretations the authors discuss. For instance, as rooted, the tree suggests Ctenophore channels belong to clade A and they authors interpret clade B has been lost in ctenophores. But what if the tree is instead rooted between ctenophores and the other animals under the assumption ctenophores, not sponges, are the outgroup? Now doesn’t this place sponge within clade B and suggest it has lost Clade A? The authors acknowledge Deg losses in freshwater sponge already, so isn’t this also a reasonable interpretation? Sponge sequence divergence could just be from reduced selection? The point is that the tree as presented is not the only reasonable interpretation based on the information the authors provide. This should be acknowledged and discussed in its entirety, or the authors could just remove the relevant conclusions they do make. Similarly, its probably not wise to wade into the ctenophore-first vs sponge-first debate with a single highly divergent gene family given the issue is contentious even with extremely large data sets.

We thank the reviewer for his comment. We now also performed a likelihood and Bayesian analysis using Ctenophora as outgroup and present it in Supplementary Figures 4 - 7. As expected by the reviewer, this analysis indeed placed sponges within clade B, sister to subclades V-VI, and suggests that they lost clade A.

3. The authors suggest a high diversity of deg/enac channels in ancestral animals both in their intro and discussion, but is this actually what the phylogeny says? My reading is it does not at all say this. Doesn’t it say there is separate lineage-specific diversification from a small set of ancestral channels? Just 2 if clades A and B hold together? And still only two present in multiple lineages of cnidarians or bilaterians? So that most diversification has occurred after divergence of the cnidarian and bilaterian classes? The diversification patterns and timing are very interesting but interpretations are not consistent with the data as presented.

Thank you for the insightful comment. We agree with this interpretation and deleted the word “ancient” from two subtitles (in Results and Discussion). Moreover, we explicitly mention the lineage-specific diversifications from a small set of ancestral channels in the Discussion.

4. The authors pose multiple evolutions for pH sensitivity based on finding pH-sensitive NeNaC channels in clades A and B, but is this the only interpretation? There is very little functional data, especially with regards to proton gating, across large swaths of the tree. Can the authors rule out that the ancestral Deg/Enac was pH gated? If so, then there might be

several isolated losses of proton gating rather than multiple evolutions.

pH-gating of the ancestral DEG/ENaC cannot be completely ruled out. However, a recent study (published as a preprint on bioRxiv) investigated the molecular determinants of proton activation of a proton-gated DEG/ENaC from *Trichoplax*, TadNaC2, in some detail and the authors concluded that the mechanisms are fundamentally different from those of bona fide ASICs, strongly suggesting independent evolution. This new study is now cited and these findings are now discussed. In addition, we mention that the molecular mechanism of proton activation needs to be investigated in more species before we can draw definite conclusions.

5. The author's genetic/behavioural data showing NeNaC2 supports cnidocyte discharge at low pH seems solid, but what is the physiological significance? Seawater has high pH and NeNaC2 does not appear to be involved in prey-dependent discharge. Most cnidarians don't have channels closely related to NeNaC2, so this seems unlikely to be a fundamental mechanism regulating cnidocyte discharge. Maybe it just needs more explanation or citations? Similarly, is the calcium-inhibition or NeNaC8 physiologically relevant given it happens in a submicromolar concentration in an osmoconformer living in seawater? The data looks good, but is this telling us anything relevant about NeNaC function in *Nematostella*? The authors should at least discuss whether these described phenomena might or might not have real physiological significance.

The first part of this comment (proton-sensitivity of NeNaC2) is related to the comments of Reviewer #1. While we find that this is an interesting question, we are afraid that answering it would be too speculative at this stage and, at this point, would not serve the readers. Regarding the Reviewer's comment "Most cnidarians don't have channels closely related to NeNaC2, so this seems unlikely to be a fundamental mechanism regulating cnidocyte discharge", we have to respectfully disagree. We can find NeNaC2 orthologs in stony corals such as *Stylophora* and other far-related sea anemones such as *Actinia* (please see Supplementary Figure 1). Moreover, we found likely orthologs in additional hexacorallian species but did not include them in the phylogeny as their gene models were incomplete. Thus, it is likely that NeNaC2 appeared in the last common ancestor of all Hexacorallia (sea anemones and stony corals) that lived approximately 500 million years ago. Hexacorallia make a significant part of Cnidaria with roughly 4300 extant species. Concerning NeNaC8, we now mention in the Results that we do not think that the calcium-inhibition is physiologically relevant. It is an artificial way to open this channel and perform a basic biophysical analysis.

6. The conclusion that NeNaC2 doesn't significantly conduct calcium appears at odds with it having a calcium permeability almost as high as sodium and far higher than potassium. Is this interpretation based solely on native calcium activated chloride currents and the assumption that EGTA will chelate them effectively? And that protons released from EGTA will not influence results? Some sort of control for EGTA effectiveness in blocking the native current could help (and maybe use Na-EGTA or BAPTA?) Or is their literature to cite that this is an effective method? Regardless, the high calcium permeability suggests the channel contributes to calcium signaling in vivo, no?

Thank you for the comment. Referee #2 made a similar comment. We indeed made a mistake when calculating relative permeabilities. We recalculated them and found that P_{Na}/P_{Ca} is indeed 0.2, which confirms a relatively low Ca^{2+} permeability of NeNaC2.

We titrated the pH of the EGTA solution, which was injected, to pH 7.4 and considered the release of protons from EGTA therefore as negligible. The pH of the EGTA solution is now indicated in the methods.

In addition, we added a citation to a paper, in which we efficiently inhibited activation of the CaCC by injection of EGTA in HyNaC-expressing oocytes.

Minor points

1. The authors mention that Deg/Enacs are much more diverse on a sequence and functional level compared to voltage-gated channels and imply the sequence divergence is responsible for that functional divergence. Yet what they are really showing here is remarkable conservation of function despite high sequence divergence is it not? This might also be worth mentioning.

This point is related to the question of whether pH-gating is conserved or evolved independently. A conserved function implies that the ancient DEG/ENaC already served this function, which we do not know. Moreover, pH-gating may still serve different physiological functions. To avoid these complicated discussions, we prefer to not speculate on conserved functions of DEG/ENaCs.

2. The color schemes in Figure 1 are unnecessarily complex and make it very hard for the reader to quickly figure out what species individual clades belong to, or to understand which phyla are contributing to a clade. It would work better to use fewer colors to highlight just the key clades, or maybe only use one color/phylum and combine that with text labels for quick deciphering. For example, the text labels are very effective for differentiating HyNacs and NeNacs, but the thin branch colors are hard to figure out.

Thank you for the comment. As suggested, we reduced the number of colors in Figure 1.

3. The cloning section in the methods may be a bit too brief. Which 19 were selected? Why weren't the 10 others in the pool? Just lack of data on starts and stops? Or was the selection for some functional reason – like these are the ones they thought would work? Some were synthesized by a specified company, but the rest are not specified? Easy to fix but incongruent and somewhat incomplete as is.

We added more details to the cloning section in the methods.

4. Is there a paper to cite that the EGTA injection as done effectively chelates calcium transients in oocytes? Would the potential for intracellular acidification cause issues for Deg/Enac function? I think probably not, but maybe there is a reference to boost confidence?

We added a citation to a paper, in which we efficiently inhibited activation of the CaCC by injection of EGTA in HyNaC expressing oocytes.

5. The data comparing NeNac2 H141A to WT is not sufficient for concluding that H141A does or does influence apparent pH sensitivity or play a major role in gating. It could be explored in more detail, removed entirely (probably the best easy choice), or the authors could soften statements and say the mutant has altered gating and this needs further exploration to exactly determine how it is altered. Its too much to say it is dispensible for proton gating when the currents are a small fraction of WT currents unless the authors know that surface expression or conductance are altered in a way that could account for the differences in current size.

We agree that the available data on H141A does not allow to draw clear conclusions, but believe that it is still interesting and worth to be shared. We, therefore, rephrased the respective sentence.

6. Some of the figure legends are not complete stand alone descriptions and some minor editing would help the reader.

We did not want to repeat too much from the main text, but included a few more details.

REVIEWERS' COMMENTS:

Reviewer #2 (Remarks to the Author):

I think the changes improve the manuscript, well done.

Two of my original comments you mention in your rebuttal, but I'm confused by your answer.

- Why was NeNaC10 not tested for responses to protons? (I certainly don't think you need to test every existing chemical at all of your NeNacs.. But like NeNaC2, NeNaC10 responded to none of the other stimuli. What if NeNaC10 is a proton receptor too?

- For NeNaC2 is PNa/PCa 0.2 or 5?

Reviewer #3 (Remarks to the Author):

The original manuscript presented extensive data sets on the evolution of animal Deg/Enac channels and functional characterization of *Nematostella* (sea anemone) orthologs. The main issues that reduced enthusiasm were problematic interpretations of phylogenetic analyses and to a lesser extent biophysical analyses. The authors have substantively address every concern raised by reviewers for each topic to the extent that is reasonable to expect. The revised manuscript is greatly improved as a result. The authors made a great decision in the revised manuscript to present phylogenetic results from both ctenophore-first and sponge-first scenarios and discuss how that changes interpretation. They are able to show the most important features of the trees remain constant and now correctly state that the trees provide evidence for two ancestral channels with extensive lineage-specific diversifications. The additional data, presentation of alternate or competing hypotheses throughout and correction of several experimental interpretations greatly improve the paper and its potential impact.